# GPR146 in adipose tissue drives adipose-liver crosstalk and promotes hepatic steatosis in mice

Yu Shi[1,2,3], Kai Yan Cheng[1,2,3], Thi Tun Thi[1,2,3], Yifan Wang[1,2,3], Yang Yang[1,2,3], Xiaoyun Cao[1,2,3], Vanna Chhay[1,2,3], Yujia Shen[4,5], Yuchen He[6], Tianyun Zhao[7], Yan Ting Lim[7], Amy Deik [8], Courtney Dennis[8], Kerry Pierce[8], Kevin Bullock[8], Martin Wabitsch [9], Clary B. Clish [8], Alexander S. Banks [6], Radoslaw M. Sobota [7], Chad A. Cowan[10] & Haojie Yu [1,2,3] ✉

The limited therapeutic options for metabolic dysfunction-associated steatotic liver disease (MASLD) underscore the need for deeper mechanistic insight and new treatment strategies. Here, we identify the orphan G protein-coupled receptor GPR146 as a regulator of hepatic steatosis through adipose–liver crosstalk. Human genetic analyses link the GPR146 locus to circulating markers of liver injury and inflammation. In mice, both constitutive and acute GPR146 depletion protect against diet-induced obesity and hepatic steatosis. Notably, adipose-specific, but not liver-specific, GPR146 deletion reduces hepatic lipid accumulation by limiting free fatty acid (FFA) influx. Mechanistically, GPR146 promotes adipogenesis in preadipocytes via Gαq-PKC-AKT signaling, increasing lipid storage capacity, and enhances lipolysis in mature adipocytes through ERK activation, elevating circulating FFA. Together, these coordinated actions increase FFA delivery to the liver, promoting triglyceride accumulation. Our findings establish GPR146 as a pleiotropic regulator of adipose tissue biology and a potential therapeutic target for MASLD.

Metabolic dysfunction-associated steatotic liver disease (MASLD) and its sequalae are emerging health issues, affecting ~25% of the adult population worldwide[1]. MASLD comprises a spectrum of hepatic manifestations ranging from the initial steatosis to the more aggressive Metabolic dysfunction-associated steatohepatitis (MASH). MASH is characterized by chronic inflammation and development of fibrosis, which may progress further to cirrhosis, liver failure or hepatocellular carcinoma, resulting a higher demand for liver transplantation[2,3]. Although multiple therapeutic strategies targeting different pathological processes of MASH, including lipid accumulation, inflammation, and fibrosis, are under clinical trials, only two pharmacologic treatment, Resmetirom, a thyroid hormone receptor-β agonist, and Semaglutide, a GLP-1 receptor agonist, have been approved by the FDA for clinical use[4–6]. This highlights the urgent need to gain deeper understanding of pathogenesis of MASH for the future development of new therapies.

[1]Precision Medicine Translational Research Programme, Yong Loo Lin School of Medicine, National University of Singapore, Singapore, Singapore. [2]Cardiovascular- Metabolic Disease Translational Research Programme, Yong Loo Lin School of Medicine, National University of Singapore, Singapore, Singapore. [3]Department of Biochemistry, Yong Loo Lin School of Medicine, National University of Singapore, Singapore, Singapore. [4]Department of Medicine, Yong Loo Lin School of Medicine, National University of Singapore, Singapore, Singapore. [5]Cancer Science Institute of Singapore, National University of Singapore, Singapore, Singapore. [6]Division of Endocrinology, Diabetes and Metabolism, Beth Israel Deaconess Medical Center and Harvard Medical School, Boston, MA, USA. [7]Institute of Molecular and Cell Biology (IMCB), Agency for Science, Technology and Research (A*STAR), 61 Biopolis Drive, Proteos, Singapore 138673, Singapore. [8]Broad Institute of Massachusetts Institute of Technology and Harvard, Cambridge, MA, USA. [9]Pediatrics and Adolescent Medicine, Ulm University Hospital, Ulm 89075, Germany. [10]Harvard Stem Cell Institute, Harvard Medical School, Boston, MA, USA. ✉e-mail: bchhaoy@nus.edu.sg

It is now well recognized that obesity and obesity related metabolic disorders such as Type2 Diabetes Mellitus (T2DM) are major risk factors for MASLD and MASH[7]. In populations with obesity, the prevalence of MASLD varies from 60 to 95%[7,8]. A key pathological feature linking obesity to MASLD is adipose tissue dysfunction. While adipose tissue serves as the primary lipid reservoir and plays a crucial endocrine role in systemic energy balance, chronic nutrient overload can lead to a breakdown in adipose expandability, promoting lipolysis, inflammation, and release of free fatty acids (FFAs) into circulation. These excess FFAs are delivered to the liver, where they contribute to hepatic triglyceride accumulation, a hallmark of steatosis, and create a pro-inflammatory environment that promotes progression toward fibrosis[9].

Among the molecular pathways orchestrating adipose tissue function, G protein–coupled receptors (GPCRs) play a central role in regulating lipid metabolism, adipogenesis, and systemic energy balance[10]. Notably, GPCRs, such as the cannabinoid receptor CB1, have been shown to promote hepatic steatosis by increasing adipocyte lipogenesis and FFAs released into circulation. Conversely, adipose-specific deletion of CB1 protects against hepatic lipid accumulation[9,11]. These findings highlight the essential role of GPCR-mediated adipose signaling in coordinating systemic lipid flux and position adipose-resident GPCRs as important contributors to MASLD pathophysiology.

Building on this concept, we previously identified the orphan GPR146 as a regulator of hypercholesterolemia and atherosclerosis[12]. In the current study, we extend our investigation to explore the role of GPR146 in the development of obesity and MASLD. We demonstrated that both constitutive and AAV-mediated acute depletion of GPR146 protected mice from diet–induced obesity and hepatic steatosis. These metabolic benefits were sex-specific, with energy expenditure elevated in both male and female knockout mice. The effect was more pronounced in females, driven by increased UCP1-mediated thermogenesis, whereas the mechanism in males remains less clear. To delineate the tissue-specific contributions, we generated adipose- and liver-specific knockout models. Our results show that the reduction in hepatic steatosis, diet-induced obesity, and circulating FFA is primarily attributed to GPR146 deficiency in adipose tissue, which limits free fatty acid flux to the liver. Mechanistically, GPR146 promotes adipogenesis through Gαq-PKC-AKT signaling in preadipocytes and enhances lipolytic activity in mature adipocytes via ERK signaling. Together, these findings highlight a critical role for adipose GPR146 in mediating adipose-liver crosstalk and support its potential as a therapeutic target for obesity-related liver disease.

## Results

### GPR146 locus is genetically associated with markers of liver injury and inflammation

Previous GWAS studies have identified the association between *GPR146* and plasma cholesterol levels[12–15]. Subsequent analysis using the UK Biobank dataset not only confirmed this finding but also revealed an association between the causal SNP in the *GPR146* locus, rs1997243, and C-reactive protein (CRP) levels, as well as plasma liver enzymes, including gamma glutamyl-transferase (GGT), aspartate transaminase (AST), and alkaline phosphatase (ALP)[16,17]. Consistently, meta-analysis of GWAS studies from Common Metabolic Diseases Knowledge Portal[18] shows that rs1997243-G allele is linked to increased plasma levels of LDL-C, CRP, GGT, ALP, AST, and alanine aminotransferase (ALT) in humans (Supplementary Fig. 1a, b). In addition, rs1997243-G is associated with an increased expression of *GPR146* in human blood and white adipose tissue based on eQTL studies using 670 human blood samples and 591 human subcutaneous adipose tissue samples from GTEx database (Supplementary Fig. 1c, d). Collectively, these findings suggest that GPR146 may play a role in regulating liver function in humans.

### GPR146 promotes diet-induced lipid accumulation and inflammatory signaling in liver

Hepatic injury and inflammation can be triggered by various factors, including metabolic disorders such as MASLD. In some cases, MASLD can progress to a more severe form known as MASH, characterized by liver inflammation and varying degrees of fibrosis[1]. In this study, we utilized *Gpr146* whole-body knockout (*Gpr146-/-*) and floxed (*Gpr146fl/fl*) mouse models (Supplementary Fig. 2a–c) to investigate the role of GPR146 in the development of MASLD and its associated hepatic injury.

In line with human genetic findings linking elevated *GPR146* expression to increased liver enzymes indicative of liver injury, we found that *Gpr146* deficiency markedly reduced hepatic triglycerides (TG) contents in both male and female mice fed a high fat diet (HFD) (Fig. 1a, b, d, e). Additionally, serum levels of ALT were significantly decreased in GPR146-deficient mice compared to control littermates (Fig. 1c, f). Moreover, the liver weight of male GPR146-deficient mice was lower than that of their wild-type control littermates when fed HFD, whereas no significant difference was observed in female mice (Supplementary Fig. 2d).

To gain deeper insights into the underlying metabolic alterations, we performed untargeted metabolomic analysis on liver from HFD fed *Gpr146+/+* and *Gpr146-/-* mice. The most notable differences were observed in metabolites related to lipid, amino acids, and glucose metabolism. Among common lipid species, triacylglycerols (TAG) and diacylglycerols (DAG) levels exhibited the most significant reduction, followed by a trend of decreased monoacylglycrols (MAG) and lysophosphatidylethanolamine (LPE) due to GPR146 deficiency (Fig. 1g, h). Additionally, several hepatic free fatty acid (FFA) and oxidized lipid species were significantly diminished in the livers of GPR146-deficient mice (Fig. 1i). Many of these metabolites, such as palmitate, linoleate, 13-hydroxyl-octadecadienoic acid (13-HODE), and dimethylguanidino valeric acid (DMGV), have previously been reported to be upregulated in liver of MASLD patients[19,20]. Consistently, total hepatic FFA levels were significantly reduced in HFD fed GPR146-deficient mice compared to control littermates (Fig. 1j). Aside from assessing hepatic FFA levels, we also measured their concentrations in plasma. We found that both male and female GPR146-deficient mice exhibited significantly reduced circulating FFA levels compared to control littermates (Fig. 1k).

To investigate the transcriptional basis for the observed phenotypes, we performed transcriptomic analysis of liver tissue from 16h-fasted HFD-fed *Gpr146-/-* and wild-type littermates. Gene set enrichment analysis (GSEA) revealed that differentially expressed genes (DEGs) were significantly enriched in pathways related to apoptosis, inflammation, and immune response (Fig. 1l, m). Notably, multiple immune-related pathways, including TNFα signaling via NF-κB and interferon responses, were significantly downregulated in *Gpr146-/-* livers, in line with the reduced hepatic steatosis and lower circulating ALT levels (Fig. 1l, m).

In addition to downregulation of injury and inflammation-related pathways, GSEA also revealed suppression of adipogenesis-related pathways in the liver of GPR146-deficient mice, suggesting that GPR146 may promote hepatic steatosis via PPARγ-mediated transcriptional programs (Fig. 1l, m). Adipogenic hepatic steatosis has been previously linked to overexpression of PPARγ[21–24]. In line with these reports, we found that hepatic *Pparg* expression was drastically elevated by HFD feeding and significantly reduced in GPR146-deficient mice at both mRNA and protein levels (Supplementary Fig. 2e, f, g, h).

Downstream targets of PPARγ involved in lipid droplet formation and lipid uptake, including *Cidea*, *Plin4*, *Cidec*, *Cd36*, were also significantly downregulated in the livers of *Gpr146-/-* mice (Supplementary Fig. 2h). Moreover, genes associated with inflammation and fibrosis (*Tgfb1*, *Tgfbi*, *Col1a1*, *Col1a2*), and extracellular matrix (ECM) remodeling (*Dcn*, *Bgn*, *Ecm1*, *Lum*) were likewise diminished,

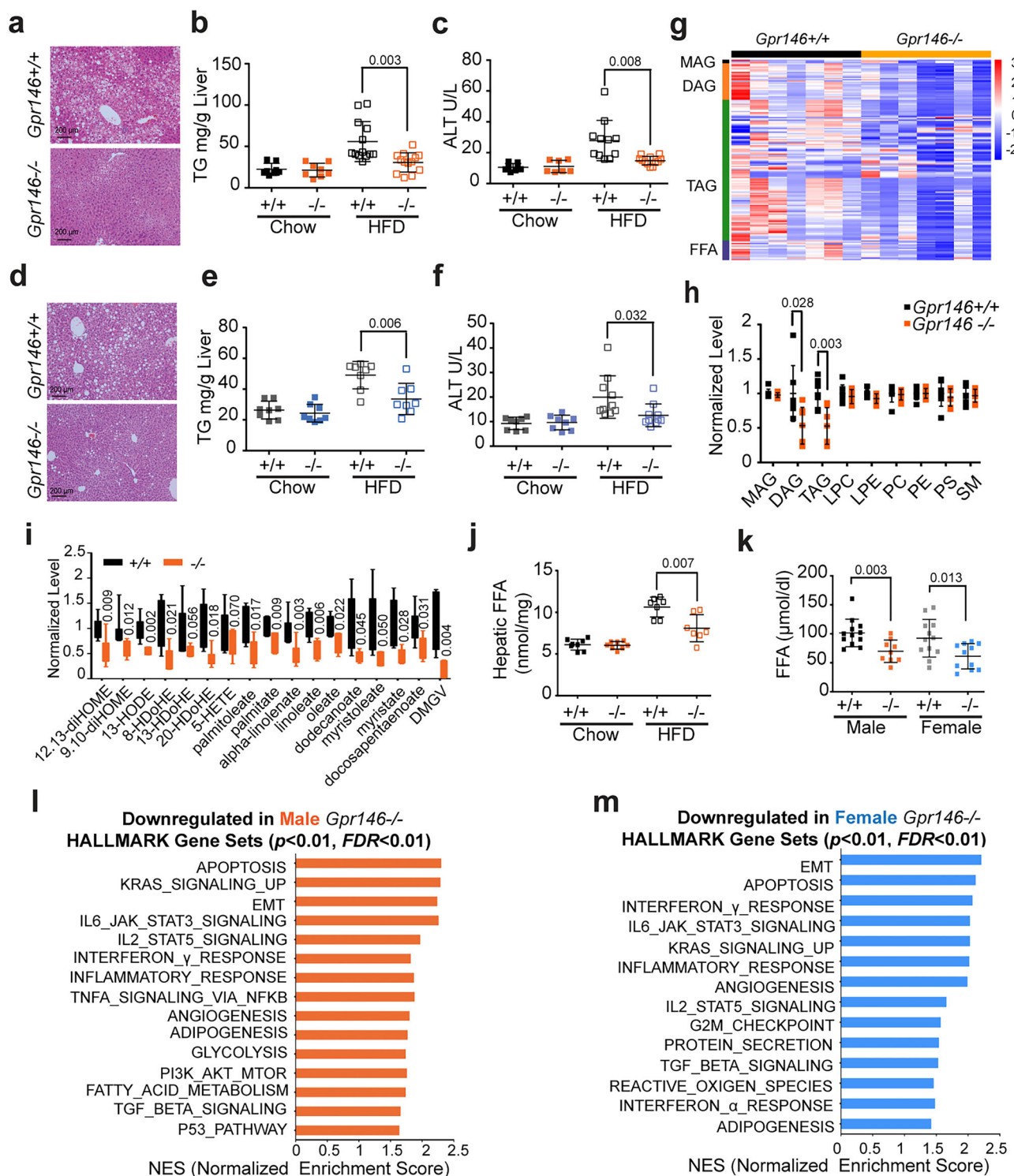

suggesting a broader suppression of MASH-associated transcriptional programs in the absence of GPR146 (Supplementary Fig. 2h). Although pathways of de novo lipogenesis and fatty acid oxidation were not significantly enriched in the transcriptomic datasets, qPCR analysis revealed modest but consistent increases in lipogenic genes (*Fasn, Acaca, Elovl6*) and reductions in fatty acid oxidation genes (*Cpt1a, Acadl*) in GPR146-deficient livers (Supplementary Fig. 2i). These changes likely represent compensatory responses to the overall reduction in hepatic lipid content. Together, these findings reveal that global GPR146 deficiency leads to suppression of hepatic PPARγ signaling and its downstream adipogenic targets, along with reduced

expression of inflammatory and ECM remodeling genes. While these changes are consistent with the reduced hepatic lipid accumulation observed in *Gpr146-/-* mice, it remains unclear whether suppression of the PPARγ pathway is a driving mechanism or a secondary adaptation to the decreased hepatic triglyceride content.

In summary, through integrated lipidomic, transcriptomic, and histological analyses, our data demonstrate that GPR146 deficiency robustly protects against diet-induced hepatic steatosis. These protective effects are associated with reduced hepatic and circulating free fatty acid levels, downregulation of inflammatory and immune response pathways, and suppression of hepatic PPARγ signaling.

**Fig. 1 | GPR146 Promotes Liver Steatosis in HFD-fed Mice.** Haematoxylin and eosin (H&E)-stained liver sections from male (**a**) and female (**d**) mice fed HFD for 3 months. Hepatic triglyceride content of 16 h fasted male (**b**) and female (**e**) *Gpr146* wild-type (+/+) mice and knockout (−/−) littermates fed chow or HFD as indicated (*n* = 8–13 mice per group). Plasma ALT levels of 16 h fasted male (**c**) and female (**f**) *Gpr146* wild-type (+/+) mice and knockout (−/−) littermates fed chow or HFD as indicated (*n* = 8–13 mice per group). **g** Heatmap of hepatic monoacylglycerol (MAG), diacylglycerol (DAG), triacylglycerol (TAG), and free fatty acid (FFA) species measured by lipidomics of 16 h fasted male mice fed HFD (*n* = 7 mice per group). **h** Hepatic total MAG, DAG, TAG, LPC (Lysophosphatidylcholine), LPE (Lysophosphatidylethanolamine), PC (Phosphatidylcholine), PE (Phosphatidylethanolamine), PS (Phosphatidylserine) and SM (Sphingomyelin) of 16 h fasted male mice fed HFD (*n* = 7 mice per group). **i** Relative level of various oxidized lipid species and FFAs in liver of male mice fed HFD for 3 months (*n* = 7 mice per group). Box plots show the median (center line), interquartile range (25th–75th percentiles; box bounds), and whiskers indicate the minimum and maximum values; individual data points represent single mice. **j** Hepatic total FFA content of 16 h fasted male *Gpr146* wild-type mice and knockout littermates fed chow or HFD as indicated (*n* = 8–13 mice per group). **k** Plasma FFA levels of 16 h fasted male and female *Gpr146* wild-type mice and knockout littermates fed HFD for 2 months (*n* = 8–13 mice per group). Top ranking HALLMARK pathway gene sets discovered from gene set enrichment analysis (GSEA) of genes differentially expressed in liver of 16 h fasted male (**l**) and female (**m**) mice, with significance assessed by permutation-based testing and FDR correction. Bars in (**b**, **c**, **e**, **f** and **h–k**) indicate mean ± s.d. Bars in (**b**, **c**, **e**, **f** and **h–k**) indicate mean ± s.d. Statistical analyses were performed using two-sided unpaired t tests (**b**, **c**, **e–k**), with P values indicated. Source data are provided as a Source Data file.

## GPR146 deficiency reprograms hepatic glucose and amino acid metabolism

Beyond reduced hepatic lipid accumulation, GPR146 deficiency led to significantly elevated levels of hexose metabolites within the pentose phosphate pathway (PPP), including hexose monophosphate and erythrose-4-phosphate (Supplementary Fig. 3a). These changes suggest a shift in glucose metabolism toward the PPP, potentially increasing NADPH production to maintain redox homeostasis[25]. We also observed elevated liver glycogen content in both male and female *Gpr146*−/− mice on chow diet, and in male knockout mice on HFD (Supplementary Fig. 3b, c). Although it remains unclear whether the glycogen accumulation reflects increased glucose uptake or reduced oxidation, these findings suggest a shift in hepatic carbohydrate flux toward storage and anabolic processing, potentially reflecting enhanced hepatic insulin sensitivity in the absence of GPR146.

Strikingly, untargeted metabolomics also revealed significantly elevated hepatic levels of seven standard amino acids in *Gpr146*−/− mice, namely histidine, cysteine, lysine, valine, leucine, isoleucine, and phenylalanine (Supplementary Fig. 3d). This pattern suggests altered amino acid handling, potentially reflecting reduced catabolism or diminished routing into gluconeogenic and lipogenic pathways[26].

Consistent with these hepatic metabolic changes, chow-fed *Gpr146*−/− mice exhibited a significantly higher respiratory exchange ratio (RER) during the dark cycle compared to controls (Supplementary Fig. 3e, f), indicating a greater reliance on glucose as an energy source. Furthermore, chow-fed *Gpr146*−/− mice displayed significantly increased lean mass without changes in fat mass (Supplementary Fig. 3g, h), potentially reflecting altered systemic nutrient partitioning and increased availability of amino acids.

In summary, GPR146 deficiency promotes a coordinated metabolic reprogramming of hepatic and systemic metabolism. These findings point to a coordinated shift in hepatic nutrient partitioning that may contribute to the reduced lipid accumulation in the absence of GPR146.

## GPR146 deficiency protects against diet-induced obesity in mice

MASLD is often accompanied with obesity and insulin resistance[1]. Starting from 3 weeks of HFD feeding, male *Gpr146*−/− mice exhibited significantly lower body weight as compared to wild-type control littermates, and this weight divergence continued to increase over time (Fig. 2a). Likewise, the body weight difference between female *Gpr146*−/− and *Gpr146*+/+ mice became discernable from 5 weeks after HFD feeding and was more pronounced as HFD feeding continued (Fig. 2e). After three months of HFD feeding, magnetic resonance imaging (MRI) analysis revealed that *Gpr146*−/− male mice had significant less fat mass than wild type controls (Fig. 2b). Both male and female GPR146-deficient mice exhibited reduced weights of inguinal (iWAT) and epididymal (eWAT) adipose tissue in comparison to their wild type controls (Fig. 2c, d, f). In male *Gpr146*−/− mice, the interscapular brown adipose tissue (BAT) was also significantly smaller, as evidenced by a reduction in depot weight (Fig. 2d). Histological analyses further showed that adipocytes in both eWAT and iWAT were markedly smaller in *Gpr146*−/− mice (Fig. 2g, h).

In parallel with the observed reduction in adipose depot mass and adipocyte size, transcriptomic profiling of adipose tissue revealed significant metabolic reprogramming in *Gpr146*−/− mice. Gene set enrichment analysis (GSEA) showed upregulation of pathways related to fatty acid oxidation, the TCA cycle, and respiratory electron transport in both epididymal (eWAT) and inguinal (iWAT) fat depots, but not in brown adipose tissue (BAT) (Supplementary Fig. 4a). These changes likely reflect enhanced mitochondrial oxidative capacity and increased fatty acid utilization within adipocytes, thereby influence systemic FFA dynamics. Concurrently, immune-related pathways were downregulated (Supplementary Fig. 4b, c, d), accompanied by reduced expression of macrophage markers, including *Cd68* and *Itgam* across all three adipose depots (Supplementary Fig. 4a). Consistent with these transcriptomic changes, histological analysis showed a marked reduction in crown-like structures (CLS) in the eWAT of *Gpr146*−/− mice, indicating reduced macrophage infiltration and attenuated adipose tissue inflammation (Fig. 2i, j).

We next assessed whether GPR146 deficiency influences systemic glucose metabolism. In male *Gpr146*−/− mice fed HFD, glucose tolerance was significantly improved compared to control littermates, with differences becoming most apparent during the late phase of the glucose tolerance test (≥60 min), suggesting enhanced glucose clearance capacity (Fig. 2k). Similarly, insulin tolerance tests showed divergence between genotypes primarily during the late recovery phase from insulin-induced hypoglycemia (≥30 min), rather than during the initial insulin response (Fig. 2l and Supplementary Fig. 5b). This temporal pattern-where genotype difference emerges primarily during the late phase of glucose and insulin tolerance tests suggests that GPR146 deficiency may enhance hepatic insulin sensitivity or modulate counter-regulatory responses, rather than directly improving peripheral insulin sensitivity and glucose disposal. Supporting this interpretation, fasting glucose, plasma insulin levels, and HOMA-IR were comparable between *Gpr146*−/− and *Gpr146*+/+ animals (Supplementary Fig. 5c–e).

In summary, GPR146 deficiency confers protection against diet-induced obesity, as demonstrated by reduced fat mass, smaller adipocyte size, and lower circulating free fatty acid levels. These phenotypes are accompanied by enhanced oxidative metabolism and reduced adipose tissue inflammation, highlighting a critical role for GPR146 in promoting white adipose tissue expansion and metabolic dysfunction under conditions of nutritional excess.

## GPR146-deficient mice exhibit elevated energy expenditure

To investigate whether altered energy utilization contributes to the lean phenotype of *Gpr146*−/− mice, and to distinguish potential causal mechanisms from secondary adaptations, we performed indirect calorimetry[27,28] after 2 weeks of HFD feeding, a time point when fat

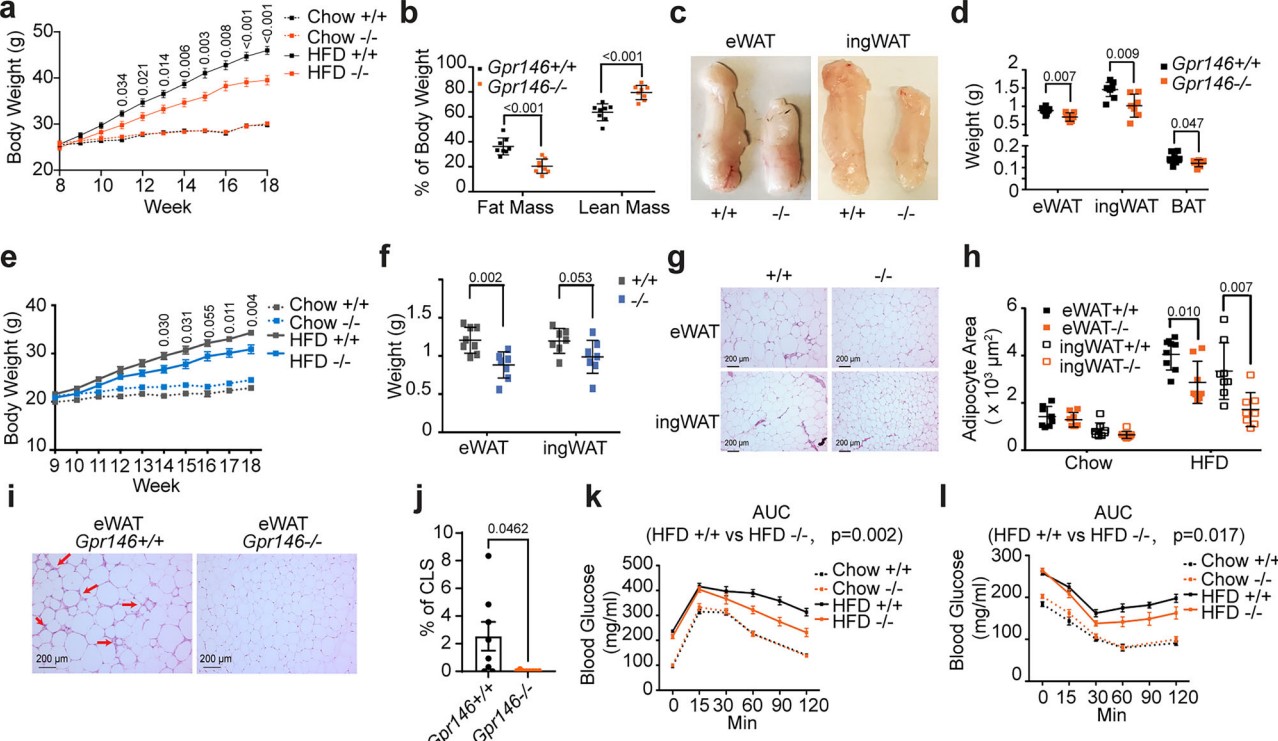

**Fig. 2 | GPR146 promotes HFD-induced obesity in mice. a** Body weight of male *Gpr146*[+/+] and *Gpr146*[-/-] littermates fed chow or HFD as indicated (*n* = 10–15 mice per group). **b** Magnetic resonance imaging (MRI) analysis of fat mass and lean mass of male *Gpr146*[+/+] and *Gpr146*[-/-] littermates fed HFD for 3 months (*n* = 8 mice per group). **c** Representative images of eWAT and ingWAT depots from male *Gpr146*[+/+] and *Gpr146*[-/-] littermates fed HFD for 3 months. **d** Weights of eWAT, ingWAT, and BAT depots from male *Gpr146*[+/+] and *Gpr146*[-/-] littermates fed HFD for 3 months (*n* = 8 mice per group). **e** Body weight of female *Gpr146*[+/+] and *Gpr146*[-/-] littermates fed chow or HFD as indicated (*n* = 9-14 mice per group). **f** Weights of eWAT and ingWAT depots from female *Gpr146*[+/+] and *Gpr146*[-/-] littermates fed HFD for 3 months (*n* = 8 mice per group). Representative images of haematoxylin and eosin (H&E)-stained eWAT and ingWAT sections (**g**) and mean area of adipocytes (**h**) from male mice fed HFD (*n* = 8 mice per group). Representative images of (H&E)-stained eWAT with Crown-like structure (CLS) indicated by red arrows (**i**) (*n* = 8 mice per group). Quantification of CLSs per 100 adipocytes (**j**). Plasma glucose during glucose tolerance test (**k**) and insulin tolerance test (**l**) (*n* = 7–10 mice per group). Bars in (**a**, **e**, **f**, **j**, **k** and **l**) indicate mean± s.e.m.; bars in (**b**, **d** and **h**) indicate mean ± s.d.. Statistical analyses were performed using two-sided unpaired t tests (**b**, **d**, **f**, **h**, **j**) or two-way ANOVA (**a**, **e**), with P values or adjusted P values indicated. For panel **k** and **l**, AUC was calculated and compared between HFD-fed *Gpr146*[+/+] and *Gpr146*[-/-] mice using two-sided unpaired t test with P values indicated. Source data are provided as a Source Data file.

mass differences between *Gpr146*[-/-] mice and their wild-type littermates begin to emerge (Fig. 3a). Food intake was comparable between genotypes (Fig. 3b, c), ruling out the possibility of excessive dietary TG contributing to the observed obesity and hepatic steatosis phenotype. Notably, male GPR146-deficient mice exhibited significantly elevated energy expenditure only during the light/inactive phase, indicating that the lack of GPR146 primarily affects resting energy homeostasis (Fig. 3d, e, Supplementary Fig. 6a–d). In contrast, female knockout mice exhibited elevated energy expenditure throughout the dark/active phase (Fig. 3g, h), indicating a potentially distinct regulatory mechanism compared to males.

The increase in energy expenditure was not attributable to changes in locomotor activity, as *Gpr146*[+/+] and *Gpr146*[-/-] littermates exhibited comparable activity levels (Fig. 3f). To explore potential mechanisms underlying the elevated energy expenditure observed in male knockout mice during the light cycle, we examined the expression of genes associated with three major thermogenic pathways: UCP1-mediated uncoupling, creatine cycling, and calcium cycling. No significant changes were detected in the expression of classical thermogenic genes (*Ucp1*, *Dio2*, *Prdm16*), creatine metabolism genes (*Ckmt1*, *Ckmt2*), or calcium cycling regulators (*Atp2a2*, *Ryr1*, *Ryr2*) in BAT, eWAT, or iWAT (Supplementary Fig. 6e–g). These results suggest that the increased energy expenditure in male *Gpr146*[-/-] mice is not driven by transcriptional activation of known thermogenic programs and may involve alternative or post-transcriptional mechanisms.

In contrast, female *Gpr146*[-/-] mice showed robust activation of classical thermogenic program. Specifically, *Ucp1* expression increased more than 15-fold in eWAT, but not in iWAT, along with significant upregulation of *Pgc1a* and *Ppara* (Supplementary Fig. 6h, i). These findings suggest that, in female *Gpr146*[-/-] mice, UCP-1 mediated thermogenesis in eWAT likely contributes to the elevated energy expenditure.

Taken together, our findings suggest that the reduced body weight and adiposity observed in GPR146-deficient mice are likely due to an increase in energy expenditure. In females, this is associated with transcriptional activation of UCP1-dependent thermogenesis. In males, however, the mechanism underlying elevated energy expenditure remains unclear. This emerging sex difference in thermogenic regulation may reflect distinct hormonal influences on adipose tissue plasticity, highlighting the need for further investigation into sex-specific pathways mediating GPR146-dependent metabolic control.

## Adipose GPR146, rather than hepatic GPR146, mediates protection against diet-induced obesity and liver steatosis

To determine the tissue origin responsible for the reduced hepatic lipid accumulation observed in *Gpr146*[-/-] mice, we first generated liver-specific *Gpr146* knockout mice (*Alb-Cre*[+] *Gpr146*[fl/fl] mice) and challenged them with HFD feeding. Surprisingly, liver-specific GPR146 depletion did not reduce hepatic TG content or improve metabolic parameters. In fact, male liver-specific knockout mice exhibited

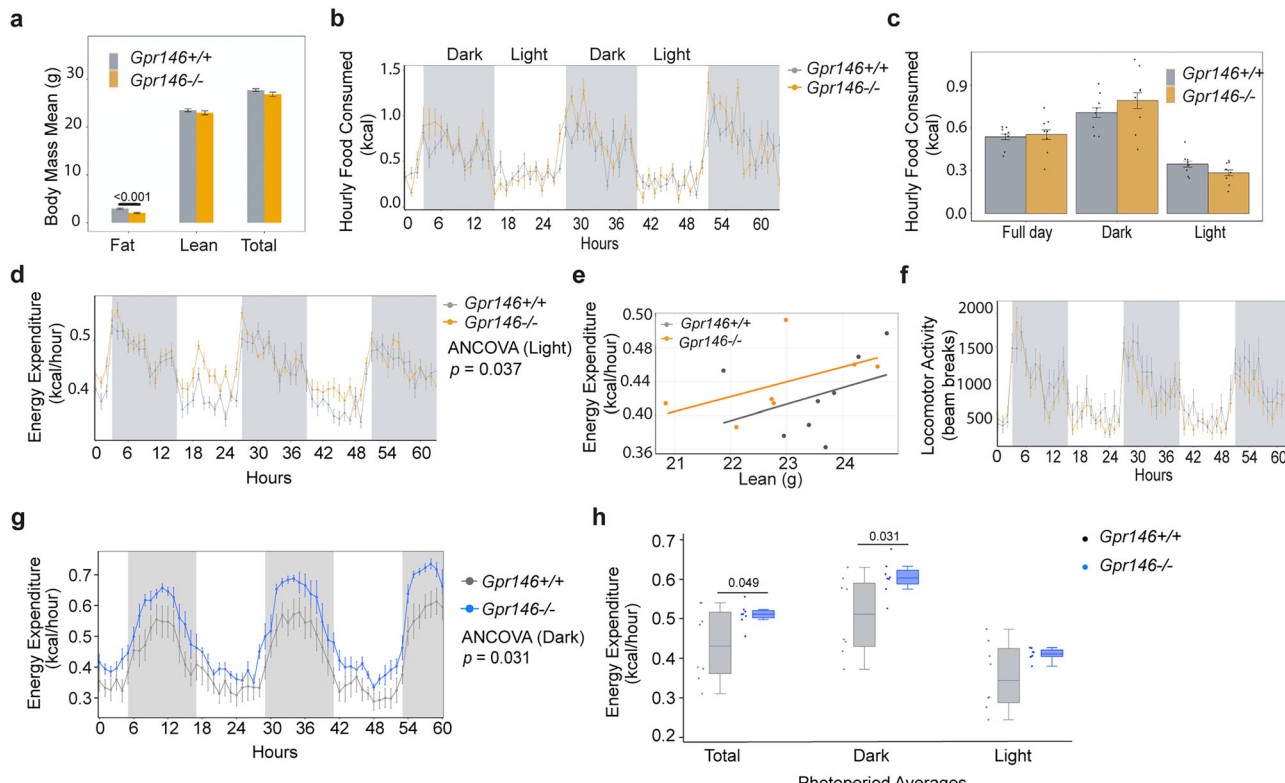

**Fig. 3 | GPR146-deficient Mice exhibit Elevated Energy Expenditure. a–f** Indirect calorimetry study in male mice fed HFD for two weeks. MRI analysis of fat mass and lean mass of male $Gpr146^{+/+}$ and $Gpr146^{-/-}$ littermates fed HFD for 2 weeks ($n = 8$ mice per group) (**a**). Hourly food intake (**b**, **c**), energy expenditure (EE) (**d**, **e**), and locomotor activity (**f**) for male $Gpr146^{+/+}$ and $Gpr146^{-/-}$ littermates monitored for 60 h and assessed using CalR software. **e** Regression plot for analysis of energy expenditure versus lean mass. **g** Indirect calorimetry study in female mice fed HFD for two weeks ($n = 8$ mice per group). Hourly energy expenditure (EE) in female $Gpr146^{+/+}$ and $Gpr146^{-/-}$ mice was recorded over 60 h ($n = 8$ mice per group). **h** Mean hourly energy expenditure (kcal/hour) per mouse during total, dark, and light phases ($n = 8$ mice per group). Box plots show the median (center line), interquartile range (25th–75th percentiles; box bounds), and whiskers indicate the minimum and maximum values; individual data points represent single mice. Bars indicate mean± s.e.m. Statistical analyses were performed using two-sided ANCOVA (**a–e**, **g** and **h**) or two-sided ANOVA (**f**) with adjusted P values indicated. Source data are provided as a Source Data file.

increased liver weight (Fig. 4a, b) and a trend of increased TG content (Fig. 4c), while female knockouts displayed significantly increased body weight under HFD feeding (Supplementary Fig. 7a). Notably, circulating FFA levels remained unchanged in both sexes (Fig. 4e, f), further distinguishing the liver-specific phenotypes from whole-body GPR146 deficiency.

Given the lack of phenotype in liver-specific knockouts, we examined tissue-specific expression patterns of GPR146. Based on the Human Protein ATLAS dataset[29] (Human Protein Atlas proteinatlas.org) and our own analysis, GPR146 is most abundantly expressed in mature adipocytes within white adipose tissue (Supplementary Fig. 8a, b). Following 8 weeks of HFD feeding, $Gpr146$ expression decreased significantly in eWAT but remained stable in iWAT and BAT (Supplementary Fig. 8c). This pattern was confirmed by independent datasets[30,31] (Supplementary Fig. 8d, e). Furthermore, transcriptomic meta-analysis of human adipose tissue from the Adipose Tissue Knowledge Portal[32] revealed consistent downregulation of GPR146 expression in individuals with obesity compared to lean controls (Supplementary Fig. 8f).

To test the functional role of adipose GPR146, we generated adipose-specific $Gpr146$ knockout mice ($Adipoq$-$Cre^+$ $Gpr146^{fl/fl}$ mice) and control littermates ($Adipoq$-$Cre^-$ $Gpr146^{fl/fl}$ mice) and subjected them to HFD challenge. Compared to control littermates, adipose-specific knockout mice exhibited significantly lower liver weight and less hepatic TG content (Fig. 4a–d). In contrast to the liver-specific knockout model, $Adipoq$-$Cre^+$ mice showed downregulation of hepatic PPARγ target genes and ECM-associated genes (Fig. 4g), paralleling the

expression profile seen in whole-body knockouts. Both male and female adipose-specific knockout mice displayed significantly lower body weight and fat mass in response to HFD challenge (Fig. 4h–j and Supplementary Fig. 7b). In males, both iWAT and BAT, but not eWAT were significantly reduced (Fig. 4k). The reduced iWAT mass was accompanied by a decrease in adipocyte size, indicating a mitigation of hypertrophy expansion (Fig. 4l). In female mice, the weight of both eWAT and iWAT depots were reduced in $Adipoq$-$Cre^+$ mice (Fig. 4m). Importantly, plasma FFA levels were significantly reduced in both sexes (Fig. 4e, f), supporting a role for adipose GPR146 in regulating systemic FFA homeostasis.

Despite these improvements, adipose-specific deletion of $Gpr146$ only partially recapitulated the phenotypes observed in whole-body knockout mice. For example, male adipose-specific knockout mice exhibited a 22% reduction in hepatic TG (Fig. 4c) and 16% reduction in fat mass (Fig. 4i), compared to 46% (Fig. 1b) and 44% (Fig. 2b) reductions in whole-body knockouts. This discrepancy may stem from depot-specific differences in gene deletion efficiency. While $Gpr146$ was substantially depleted in iWAT, its reduction in eWAT diminished over time with HFD, despite efficient recombination in chow-fed young adults (Supplementary Fig. 7c, 7d). The reason for this apparent loss of knockdown efficiency in eWAT over time is not fully understood. One possible explanation is regional variation in Cre recombination efficiency under chronic metabolic stress. Additionally, obesity is known to induce substantial changes in adipose tissue composition where there is marked infiltration of immune cells and a corresponding reduction in the proportion of mature adipocytes[33] - the specific cell

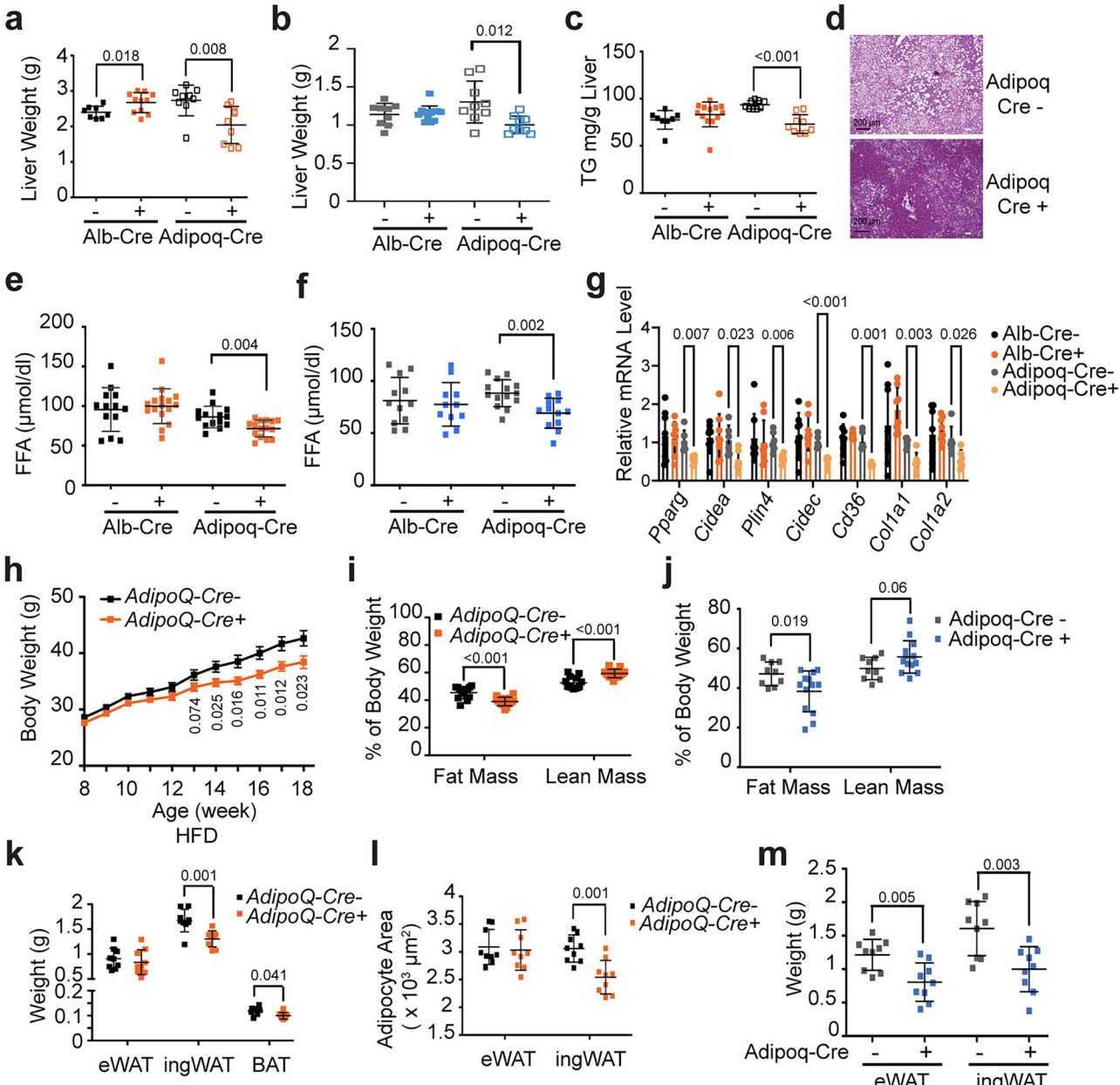

**Fig. 4 | Adipose-resident GPR146 promotes liver steatosis in HFD-fed mice.** Liver weights from male (**a**) and female (**b**) liver-specific knockout mice (Alb-Cre[+]) and control littermates (Alb-Cre[−]) fed HFD for 4 months (n = 9–13 mice per group), and from male (**a**) and female (**b**) adipose-specific knockout mice (Adipoq-Cre[+]) and control littermates (Adipoq-Cre[−]) fed HFD for 5 months (n = 8–13 mice per group). **c** Hepatic triglyceride content in male mice from the liver-specific and adipose-specific cohorts (n = 8–13 mice per group). **d** Representative H&E-stained liver sections from male adipose-specific knockout mice and control littermates fed HFD for 4 months (n = 8–13 mice per group). Plasma FFA levels of male (**e**) and female (**f**) liver-specific knockout and adipose-specific knockout mice, and corresponding control littermates fed HFD for 3 months (n = 10–15 mice per group). **g** qPCR expression analysis of hepatic adipogenesis- and extracellular matrix-related genes from male liver-specific knockout mice and control littermates fed HFD for 3 months (n = 8–13 mice per group); and from male adipose-specific knockout mice

and control littermates fed HFD for 4 months (n = 8–13 mice per group). **h** Body weight of male adipose-specific knockout mice and control littermates fed HFD as indicated (n = 10–15 mice per group). MRI analysis of fat mass and lean mass from male (**i**) and female (**j**) adipose-specific knockout mice and control littermates fed HFD fed HFD for 4 months and 5 months, respectively (n = 12–15 mice per group). **k** Weights of eWAT, ingWAT, and iBAT depots from male adipose-specific knockout mice and control littermates after 4 months HFD (n = 9 mice per group). **l**, Mean adipocyte area in eWAT and ingWAT (n = 9 mice per group). **m** Weights of eWAT and ingWAT depots from female adipose-specific knockout mice and control littermates fed HFD for 5 months (n = 9 mice per group). Bars in (**a**–**c**, **e**, **f**, **i**–**m**) indicate mean ± s.d., bars in (**g** and **h**) indicate mean ± s.e.m. Statistical significance for panels (**a**–**c** and **e**–**m**) was determined by two-sided unpaired t tests, with P values shown. Source data are provided as a Source Data file.

type targeted by Adipoq-Cre. This deficit in mature adipocytes could dilute the overall depletion effect observed at the mRNA level when analyzing bulk tissue.

We next assessed systemic consequences of adipose *Gpr146* deletion using indirect calorimetry. Unlike whole-body knockouts, adipose-

specific knockouts did not exhibit increased energy expenditure despite the significant reduction in body weight and fat mass (Supplementary Fig. 7e, f). However, they displayed a significant increase in respiratory exchange ratio (RER) during the light cycle (Supplementary Fig. 7g), indicating a shift toward carbohydrate utilization.

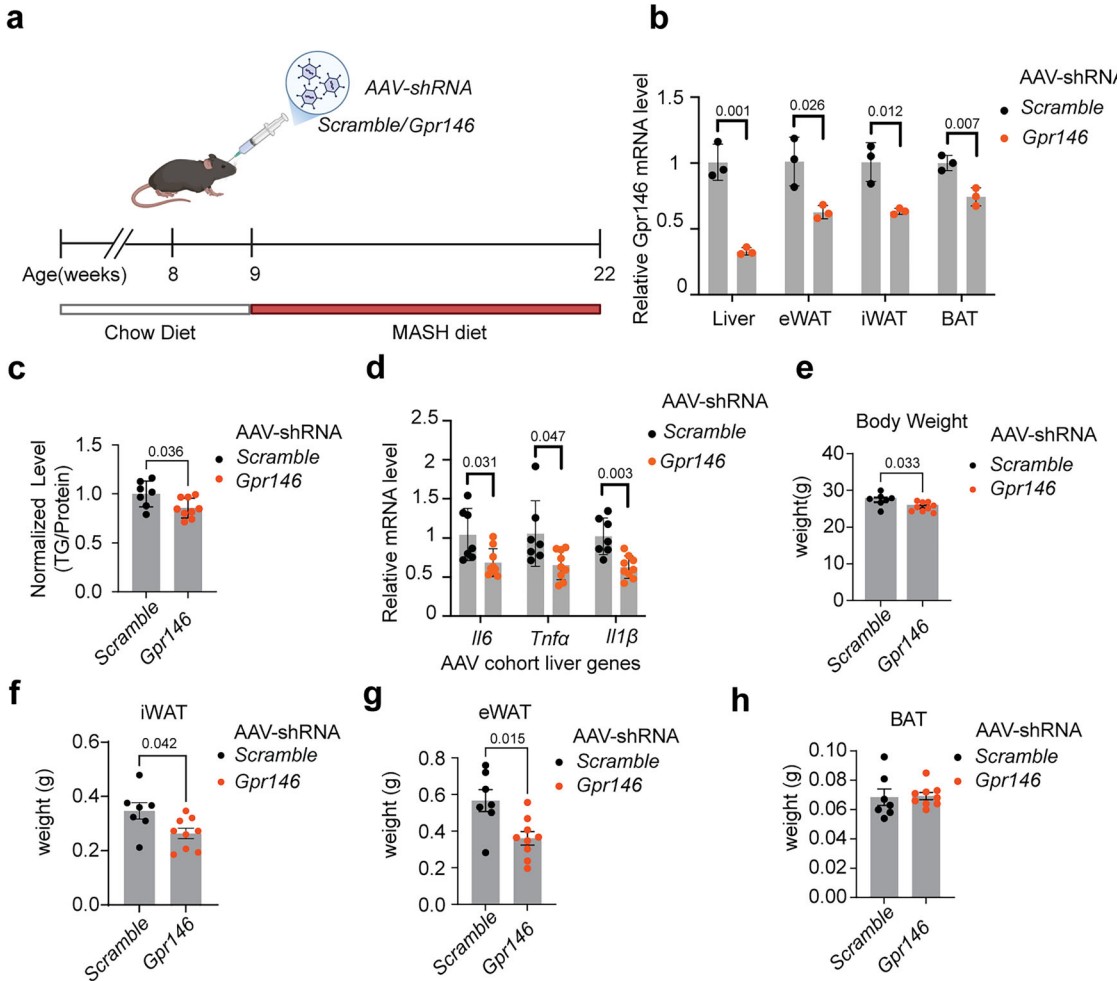

**Fig. 5 | AAV-mediated acute depletion of GPR146 confirms its role in promoting hepatic steatosis and adiposity. a** Schematic timeline of the AAV cohort. Adult male mice were injected with AAV8 expressing shRNA targeting *Gpr146* (*n* = 9) or scramble control (*n* = 7 mice per group), followed by 13 weeks of feeding with a MASH-inducing diet enriched in fat, cholesterol, and fructose. Created in BioRender. Cheng, K. Y. (2026) https://BioRender.com/8v6bx97. **b** qPCR expression analysis of *Gpr146* in liver, eWAT, iWAT, and BAT to assess knockdown efficiency (*n* = 3 mice per group). **c** Hepatic triglyceride content in mice with or without *Gpr146* knockdown (*n* = 7 mice per group). **d** qPCR expression analysis of hepatic expression of key inflammatory genes in mice with or without *Gpr146* knockdown (*n* = 7 mice per group). Body weight (**e**), iWAT(**f**), eWAT (**g**), and BAT (**h**) weights after 13 weeks of MASH diet feeding (*n* = 7 mice per group). Bars indicate mean ± s.e.m. Statistical significance for panels (**b–h**) was determined by two-sided unpaired t tests, with P values shown. Source data are provided as a Source Data file.

Taken together, these findings demonstrate that adipose-resident GPR146, rather than hepatic GPR146, is primarily responsible for the regulation of diet induced obesity, hepatic steatosis and circulating FFA levels in response to HFD.

## AAV-mediated acute knockdown of *Gpr146* confirms its role in promoting hepatic steatosis and adiposity

To further validate the regulatory role of GPR146 in hepatic lipid accumulation and to rule out the possibility that the phenotypes observed in germline knockout mice result from developmental compensation, we employed an AAV-mediated approach to acutely knock down *Gpr146* in adult mice. This strategy allows temporal control of gene silencing in fully developed animals, enabling us to assess the metabolic effects of GPR146 depletion in a mature physiological context. Following AAV-shRNA delivery, mice were fed a MASH-inducing diet rich in fat, cholesterol and fructose, which better recapitulates human liver inflammatory pathology[34] (Fig. 5a). This approach achieved robust knockdown in the liver (~70% reduction in mRNA levels), along with significant depletion in both white and brown adipose tissues (Fig. 5b). Consistent with the findings from germline knockout models, AAV-mediated GPR146 depletion significantly

reduced hepatic triglyceride content and suppressed the expression of key inflammatory markers in the liver (Fig. 5c, d).

Beyond the liver, AAV-mediated knockdown of *Gpr146* in adult mice also led to significantly lower body weight and reduced white adipose depot mass (Fig. 5e–h). The most pronounced effect was observed in eWAT, which was reduced by approximately 40% compared to control littermates (Fig. 5g). These results reinforce the role of GPR146 in promoting diet-induced obesity and liver steatosis and confirm that its metabolic effects are not limited to developmental loss of function.

## GPR146 promotes differentiation of both mouse and human preadipocytes

In response to overnutrition, adipose tissue undergoes expansion through hypertrophy (enlarged adipocytes), hyperplasia (increased numbers of adipocytes) or both[35]. Given the observed reduced fat mass and adipocyte size in both adipose-specific and whole body *Gpr146* knockout mice, we sought to investigate whether and how GPR146 regulates adipose biology.

We first isolated stromal vascular fraction (SVF) and adipocyte fraction from iWAT of *Gpr146*[+/+] and *Gpr146*[–/–] mice fed with HFD for 3

months and assessed their gene expression profiles. In SVF fraction, we observed a significant reduction in the expression of *Pparg* and its downstream gene *Fabp4*, along with decreased expression of macrophage markers *Adgre1* (*F4/80*) and *Mcp1* in the absence of GPR146 (Fig. 6a). In adipocyte fraction, genes related to transport and uptake of fatty acid (*Fabp4* and *Cd36*) were significantly suppressed upon GPR146 depletion, suggesting reduced lipid storage capacity within adipocyte (Fig. 6b). Consistent with our microarray data, we found a significant increase in the expression of *Fasn*, which encodes fatty acid synthetase, and *Ppara*, which encodes PPARα, the master transcriptional regulator for β-oxidation, in adipocytes lacking GPR146 (Fig. 6b). Collectively, these results suggest that GPR146 might regulate both hypertrophic and hyperplastic expansion of adipose tissue.

To further study the regulatory role of GPR146 in adipogenesis, we characterized the expression pattern of *GPR146* throughout the differentiation of mouse SVF and human preadipocyte SGBS cells. Notably, *GPR146* expression gradually increased over time during differentiation, peaking in mature adipocyte (Supplementary Fig. 9a, b). This observation aligns with the findings from adipose tissue single cell sequencing data showing that *GPR146* is most highly expressed in mature adipocytes (Supplementary Fig. 8b). In addition, the expression pattern of *GPR146* closely resembled that of *Pparg*, suggesting a potential role for GPR146 in regulating adipogenesis (Supplementary Fig. 9b). To test this hypothesis, we knocked down *Gpr146* expression in mouse SVF using siRNA and found that the knockdown cells exhibited significantly lower total TG content compared to control cells following differentiation into adipocytes (Fig. 6c, d and Supplementary Fig. 9c). To determine whether the reduced TG content reflected impaired commitment to adipogenesis, diminished cellular lipid accumulation, or both, we quantified the percentage of BODIPY-positive cells. GPR146 depletion significantly reduced the percentage of BODIPY-positive cells, indicating reduced adipocyte differentiation (Fig. 6e). Moreover, we employed a high-content imaging system to quantify the BODIPY-stained area of each differentiated adipocyte, serving as a proxy for TG content within individual adipocyte. Our findings indicate that the average BODIPY area of differentiated adipocytes from knockdown cells is smaller than control cells, suggesting a diminished TG accumulation in adipocytes upon GPR146 depletion (Fig. 6f, g). Gene expression analysis showed significantly lower levels of *Pparg* and *Cebpa* in *Gpr146* knockdown cells compared to control cells following differentiation (Fig. 6h). while *Cebpb*, which encodes the transcription regulator typically activated in the early stage of differentiation and responsible for adipocyte commitment, was unaffected (Fig. 6h). Therefore, these data suggest that GPR146 is more likely to regulate the later stages of adipocyte differentiation rather than the early commitment phase.

To validate these findings in a loss-of-function model, we performed ex vivo CRISPR/Cas9-mediated knockout of *Gpr146* in SVF cells isolated from Cas9-expressing mice. Similar to the siRNA data, *Gpr146* knockout significantly impaired adipocyte differentiation, as evidenced by reduced triglyceride accumulation (Supplementary Fig. 9d, e) and marked downregulation of *Cebpa* and *Pparg* (Supplementary Fig. 9f).

We next asked whether this regulatory role is conserved in human cells. Knockdown of *GPR146* in SGBS cells also impaired adipocyte differentiation, as indicated by reduced lipid accumulation and lower expression of *PPARG*, *CEBPA* along with *FASN* and *FABP4*, which are involved in the terminal differentiation of adipocyte (Fig. 6i–l and Supplementary Fig. 9g). Additionally, we assessed mitotic clonal expansion (MCE), which has been shown to support early adipogenesis in both murine and human preadipocytes, by measuring using 5-bromodeoxyuridine (BrdU) incorporation 24 h after induction of adipogenesis. GPR146 depletion significantly reduced BrdU incorporation in both SGBS and mouse SVF cells, suggesting that GPR146 regulates adipogenesis at least partially through MCE (Fig. 6m, n).

Interestingly, lentivirus-based overexpression of *GPR146* in SGBS cells did not enhance adipogenesis. Instead, it significantly impaired differentiation, as shown by reduced lipid accumulation (Supplementary Fig. 9h, i). This data suggests that GPR146 does not operate through a simple dose-dependent mechanism. Rather, its expression must be maintained within an optimal range to preserve adipogenic capacity. Collectively, these gain- and loss-of-function studies in both mouse and human cells underscore the importance of finely tuned GPR146 activity in regulating adipocyte differentiation.

Altogether, these results establish that GPR146 promotes adipocyte differentiation in preadipocytes.

## GPR146 regulates adipogenesis through Gαq-mediated activation of PKC signaling pathway in both human and mouse preadipocytes

To elucidate the molecular mechanism by which GPR146 regulates adipogenesis, we performed phosphoproteomics profiling of SGBS cells with or without *GPR146* knockdown following three days of differentiation. Pathway analysis by Metascape[36] revealed an enrichment of phosphorylated proteins associated with cytoskeleton rearrangement, fatty acid synthesis and adipogenesis, suggesting a significant impact of GPR146 depletion on adipogenesis (Fig. 7a). Dysregulated adipogenesis can be influenced by one or more protein kinases and their downstream signaling cascades. To gain further insights into the potential kinases regulated by GPR146, we next conducted kinase substrate enrichment analysis (KSEA)[37]. This approach assesses changes in kinase activities resulting from *GPR146* knockdown on the basis of previously annotated kinase substrates (PhosphositePlus) and predicted kinase-substrate relations (NetworKIN)[37]. KESA recapitulated the significant downregulation of multiple kinases involved in various pathways in *GPR146* knockdown cells, including cAMP-PKA signaling (PRKAA2, PRKACG, PRKACA, PRKAC), mTOR signaling (RPS6KB1, RPS6KA1), PI3K-AKT signaling (AKT1, AKT2, PRKDC), MAPK signaling (RAF1, MAPK8) and PKC signaling (PRKCG, PRKCA) (Fig. 7b).

To validate the findings from pathway and kinase substrate analysis, we examine the levels of PPARγ, C/EBPα, phosphorylated AKT via western blot in *GPR146* knockdown and control SGBS or mouse SVF cells. Knockdown of *GPR146* resulted in decreased PPARγ and C/EBPα levels and attenuated AKT phosphorylation in both human and mouse preadipocytes undergoing differentiation (Fig. 7c, Supplementary Fig. 10a–f), supporting impaired PI3K-AKT signaling. Given that insulin signaling activates PI3K-AKT to promote adipogenesis, in part by upregulating adipogenic transcription factors such as PPARγ[38,39], our data suggest that reduced AKT activation underlies the impaired differentiation phenotype observed in GPR146-deficient cells.

Next, we proceed to investigate how GPR146 modulates AKT activation during adipogenesis. Upon ligand binding, G protein-coupled receptors (GPCRs) activate heterotrimeric G proteins and facilitates the exchange of GTP for GDP. This process results in the dissociation of the G protein into its α and βγ subunits, which subsequently elicit a variety of downstream signaling pathways[40]. There are four major sub-families of Gα subunit (Gαs, Gαi/o, Gαq/11 and Gα12/13) that associate with GPCRs in a cell- and context-dependent manner, triggering different GPCR signaling depending on their subfamilies. Typically, Gαs stimulates the activation of adenyl cyclase and increases cAMP levels, which, in turn, leads to the activation of protein kinase A (PKA) and downstream effectors. Conversely, Gαi inhibits adenyl cyclase, resulting in lower cAMP levels[41]. To investigate whether GPR146 elicits its downstream signaling through Gαs/Gαi, we knocked down *GPR146* in SGBS cells and mouse SVF cells followed by measuring the intracellular cAMP levels at different time points during differentiation. While we observed a trend of slightly increased intracellular cAMP levels in SGBS cells without GPR146, SVF cells upon GPR146 knockdown exhibited significantly decreased cAMP levels compared to control cells (Fig. 7d and Supplementary Fig. 10g). To examine if

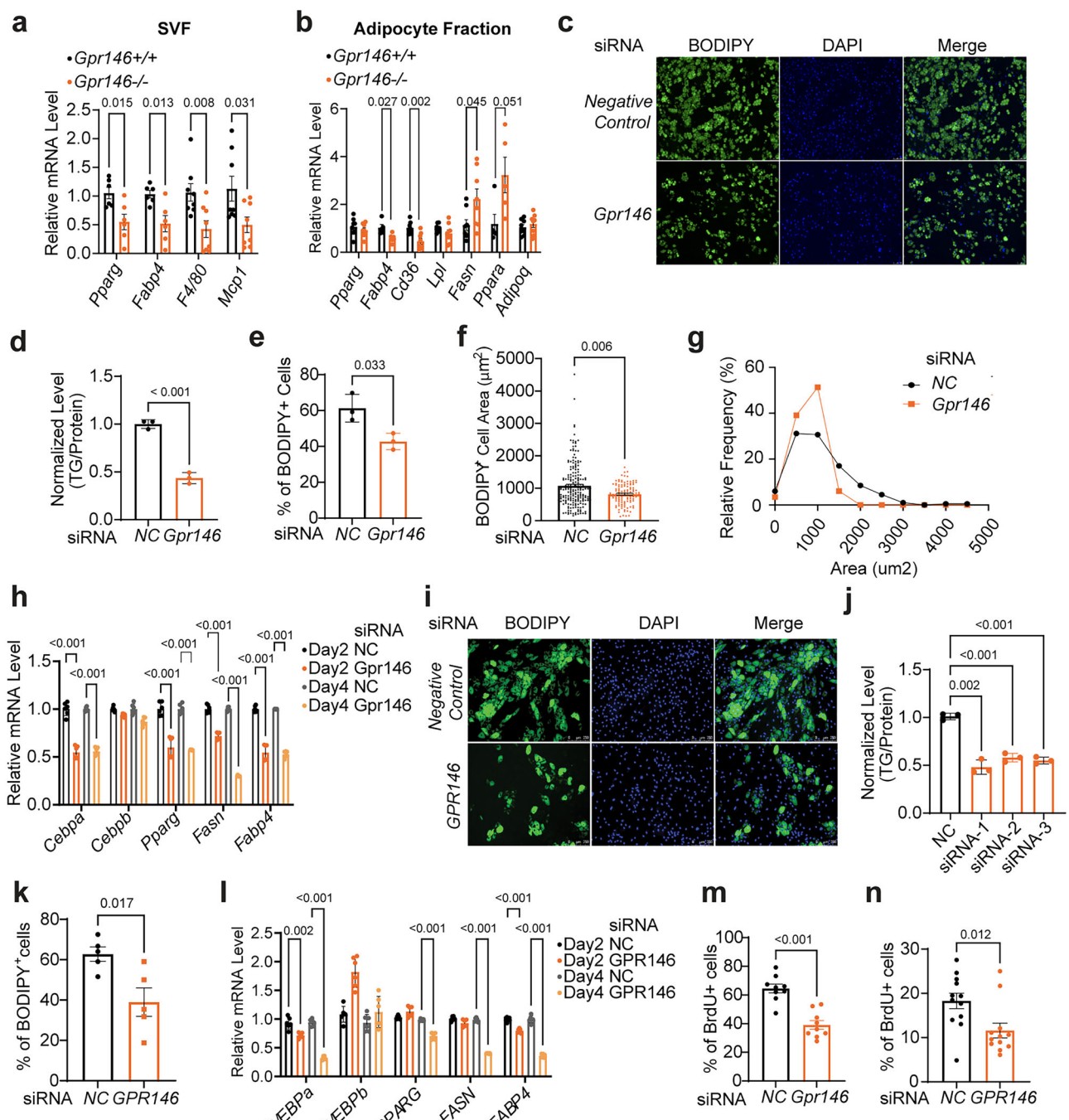

**Fig. 6 | GPR146 promotes differentiation of both mouse and human preadipocytes.** Quantitative qPCR gene expression analysis of stromal vascular fraction (**a**) and adipocyte fraction (**b**) isolated from ingWAT of male *Gpr146*⁺/⁺ and *Gpr146*⁻/⁻ littermates fed HFD for 3 months (*n* = 7 mice per group). BODIPY-staining (**c**), triglyceride (TG) content (**d**), percentage of BODIPY+ cells (**e**), average BODIPY area in adipocytes (**f**), and adipocytes area distribution (**g**) of mouse SVF-differentiated adipocytes upon siRNA-induced knockdown of *Gpr146* (*n* = 3 replicates per experiment, representative of 3 independent experiments). **h** Quantitative qPCR expression analysis of adipogenesis genes at day 2 and day 4 of differentiation in mouse preadipocytes (*n* = 3 replicates per experiment, representative of 3 independent experiments). BODIPY-staining (**i**), triglyceride (TG) content (**j**), and percentage of

BODIPY+ cells (**k**) of human primary pre-adipocyte SGBS-differentiated adipocytes upon siRNA-induced knockdown of *GPR146* (*n* = 3 replicates per experiment, representative of 3 independent experiments). **l** Quantitative qPCR expression analysis of adipogenesis genes at day 2 and day 4 of differentiation in human preadipocytes (*n* = 3 replicates per experiment, representative of 3 independent experiments). Percentage of BrdU+ cells of mouse SVF (**m**) and human SGBS (**n**) at day 4 of adipocyte differentiation upon siRNA-induced depletion of *GPR146* (*n* = 3 replicates per experiment, representative of 3 independent experiments). Bars in (**a** and **b**) indicate ± s.e.m., bars in (**d**–**f**, **h**, **j**–**n**) indicate mean± s.d. Statistical significance for panels (**a**, **b** and **d**–**h**, **j**–**n**) was determined by two-sided unpaired t tests, with P values shown. Source data are provided as a Source Data file.

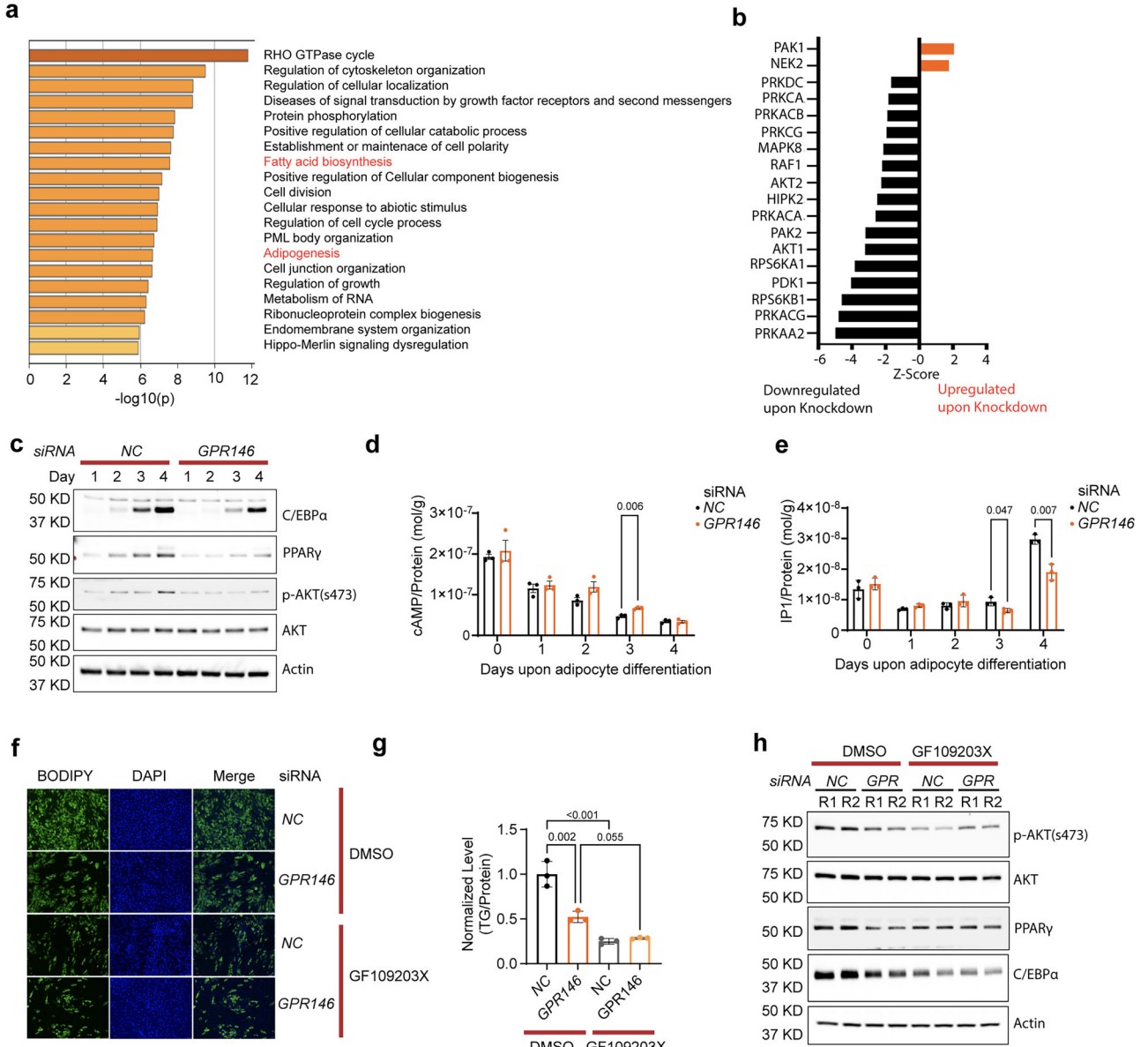

**Fig. 7 | GPR146 regulates adipogenesis through Gαq-mediated activation of PKC-AKT signaling pathway.** Proteomics and pathway analysis (**a**) Phosphoproteomics and kinase substrates analysis (**b**) of SGBS cells with siRNA-induced knockdown of *GPR146* upon adipocyte differentiation for 3 days. **c** Representative western blots of PPARγ, C/EBPα, p-AKT (s473) and ATK in *GPR146* knockdown or control SGBS pre-adipocytes upon differentiation for the time indicated (representative of 2 independent experiments). Cellular cAMP (**d**) and IP1 (**e**) levels in *GPR146* knockdown or control SGBS pre-adipocytes upon differentiation for the time indicated (n = 3 replicates per experiment, representative of 3 independent experiments). BODIPY-staining (**f**) and triglyceride (TG) content (**g**) of human primary pre-adipocyte SGBS-differentiated adipocytes upon siRNA-induced knockdown of *GPR146* in the presence or absence of GF109203X (n = 3 replicates per experiment, representative of 3 independent experiments). **h** Representative western blots of PPARγ, C/EBPα, p-AKT (s473) and ATK in GPR146 knockdown or control SGBS pre-adipocytes upon differentiation for 4 days in the presence or absence of GF109203X (representative of 2 independent experiments). Bars in (**d**, **e**, and **g**) indicate mean± s.d. Statistical analyses were performed using two-sided unpaired t tests (**d**, **e**) or two-way ANOVA (**g**), with P values or adjusted P values indicated. Source data are provided as a Source Data file.

PKA signaling regulates adipogenesis, we used PKA inhibitor H89 to treat the preadipocytes during differentiation and found that it did not affect adipogenesis and AKT signaling activity in SVF cells (Supplementary Fig. 10h–k). Therefore, these data collectively exclude the possibility of Gαs- or Gαi-mediated signaling pathways being responsible for attenuated adipogenesis in the absence of GPR146.

Given that PKC signaling pathway is also significantly downregulated in *GPR146* knockdown cells from our phosphoproteomics study, we next tested whether GPR146 regulates adipogenesis by coupling to Gαq. Gαq upon activation binds to and activates phospholipase C (PLC), which in turn cleaves phosphatidylinositol bisphosphate (PIP2) into two second messengers: diacylglycerol (DAG) and inositol triphosphate (IP3)[41]. DAG activates PKC, while IP3 triggers calcium release. We then measured cellular IP1 levels as a proxy for Gαq signaling. In SGBS cells, IP1 levels were comparable between *GPR146* knockdown and control cells up to day 2 of differentiation. However, starting from day 3 of differentiation, knockdown cells exhibited a significant reduction in IP1, with a more pronounced decrease by day 4 (Fig. 7e). A similar reduction in IP1 was observed in SVF cells at day 3 after differentiation (Supplementary Fig. 10l). Collectively, these data suggested that GPR146 likely regulates PKC signaling pathway through coupling to Gαq during adipogenesis.

To further interrogate if GPR146 regulates adipogenesis through Gαq-PKC axis, we treated SGBS and SVF cells with the PKC inhibitor GF109203X during differentiation. PKC inhibition significantly attenuated adipogenesis in both control (siNC) and GPR146 knockdown (siGPR146) cells compared to DMSO-treated counterparts. Notably, treatment with the PKC inhibitor eliminated the difference in adipogenic capacity between control and GPR146-deficient cells, suggesting that impaired PKC signaling underlies the differentiation defect caused by GPR146 loss in both human and mouse preadipocytes (Fig. 7f, g and Supplementary Fig. 10m–o). GF109203X also reduced phosphorylation of AKT and expression of PPARγ and C/EBPα in control (siControl) cells, with no further effect in knockdown(siGPR146) cells (Fig. 7h and Supplementary Fig. 10p–r). Taken together, these data suggest that GPR146 promotes adipogenesis through Gαq-mediated activation of PKC and downstream AKT signaling.

## GPR146 regulates mature adipocyte lipolysis through ERK signaling

Given the high expression of GPR146 in mature adipocytes, we next explored whether GPR146 influences adipocyte function beyond differentiation. Transcriptomic profiling of eWAT and iWAT from whole-body knockout mice revealed no significant changes in the expression of major adipokines, including *Adipoq* (adiponectin), *Lep* (leptin), *Retn* (resistin), and *Cfd* (adipsin) (Supplementary Fig. 11a, b), suggesting that GPR146 does not broadly impact endocrine output. Having established that GPR146 regulates adipogenesis through the Gαq-PKC-AKT signaling axis in preadipocytes, we next investigated whether this axis is still active in mature adipocytes. To test this, we first differentiate the SGBS cells into mature adipocytes, then knocked down *GPR146* using siRNA (Fig. 8a and Supplementary Fig. 11c). Interestingly, unlike in

preadipocytes, GPR146 knockdown in mature adipocytes did not alter the expression of *PPARG* and *CEBPA*, the key transcriptional regulators of adipogenesis (Supplementary Fig. 11c), nor did it affect insulin-stimulated AKT phosphorylation (p-AKT) (Supplementary Fig. 11d, e). These findings suggest that the GPR146-Gαq-PKC-AKT axis is primarily active during adipocyte differentiation, but not in mature adipocytes.

We then examined if GPR146 regulates lipolysis in mature adipocytes. Knockdown of *GPR146* via siRNA significantly reduced forskolin-stimulated glycerol release, indicating impaired lipolytic activity (Fig. 8b). Conversely, doxycycline (DOX)-induced overexpression of GPR146 significantly enhanced glycerol release (Fig. 8c). Notably, GPR146 knockdown did not alter the mRNA levels of the three key lipases involved in lipolysis, *LIPE* (HSL), *PNPLA2* (ATGL), and *MGLL* (MGL) (Supplementary Fig. 11c), suggesting that its regulation of lipolysis operates at the signaling level rather than through transcriptional control of these enzymes. Further mechanistic analysis revealed that ERK signaling activity was significantly decreased upon *GPR146* knockdown and increased upon overexpression in mature adipocytes (Fig. 8d and Supplementary Fig. 11f–h). Importantly, pharmacological inhibition of ERK using PD98059 abolished the lipolysis-enhancing effect of GPR146 overexpression (Fig. 8c), supporting a model in which GPR146 promotes lipolysis through ERK-dependent mechanisms. To validate these findings in vivo, we assessed lipolysis in AAV-shRNA-mediated *Gpr146* knockdown mice following stimulation with CL316,243, a β3-adrenergic receptor agonist. While CL316,243 induced modest lipolysis in control mice, *Gpr146* knockdown significantly blunted glycerol release at 30 min post-treatment (Fig. 8e), consistent with our in vitro results.

In summary, these findings demonstrate that GPR146 regulates adipose function in a stage-specific manner-promoting adipogenesis

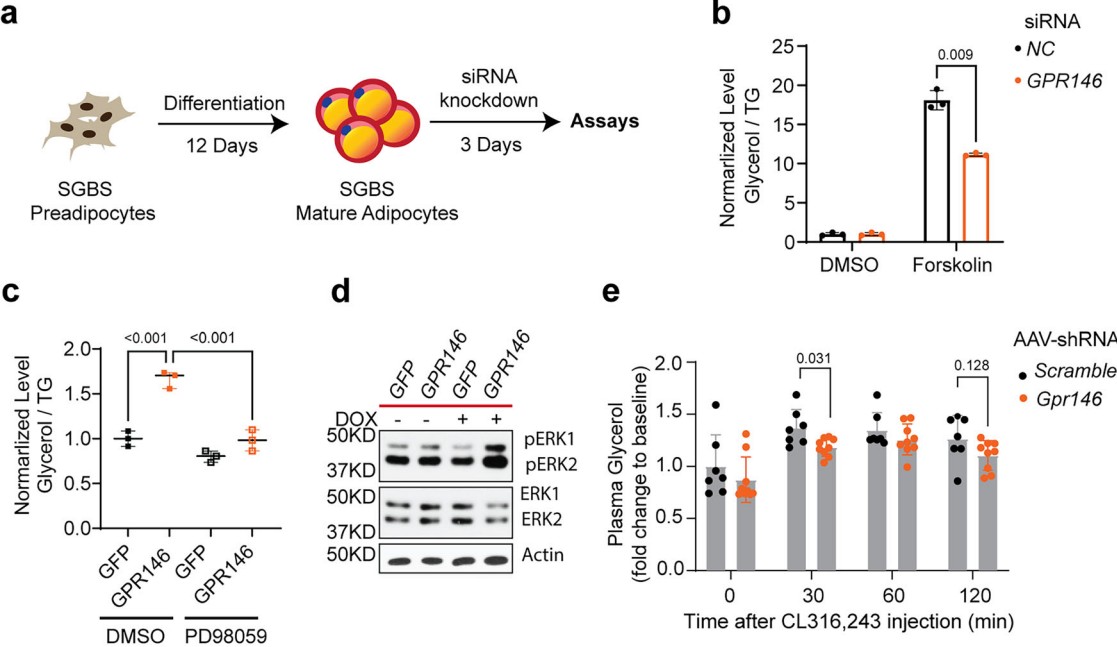

**Fig. 8 | GPR146 regulates mature adipocyte lipolysis through ERK signaling.**
**a** Workflow schematic of in vitro lipolysis performed in SGBS mature adipocytes 3 days after transfection with siRNA targeting *GPR146*. Created in BioRender. Cheng, K. Y. (2026) https://BioRender.com/9j3omk0. **b** Glycerol release from SGBS mature adipocytes transfected with siGPR146 or control siRNA, with or without forskolin stimulation; values normalized to intracellular triglyceride (*n* = 3 replicates per experiment, representative of 3 independent experiments). **c**, Glycerol release from SGBS adipocytes with doxycycline-inducible GPR146 expression under basal condition. Measurements were performed with or without the MEK inhibitor PD98059 and normalized to intracellular TG (*n* = 3 replicates per

experiment, representative of 3 independent experiments). **d** Representative Western blots of phosphorylated ERK (p-ERK) and total ERK in SGBS cells with or without doxycycline-induced GPR146 expression. **e** In vivo lipolysis assay in mice with acute depletion of *Gpr146* using AAV-shRNA or scramble control, maintained on a MASH-inducing diet for 6 weeks. Mice were injected with CL-316,243 to stimulate lipolysis, and plasma glycerol levels were measured at the indicated time points. (*n* = 7–9 mice per group). Bars indicate mean ± s.e.m. Statistical analyses were performed using two-sided unpaired t tests (**b**) or two-way ANOVA (**c**, **e**), with P values or adjusted P values indicated. Source data are provided as a Source Data file.

via Gαq-PKC-AKT signaling during differentiation and stimulating lipolysis via ERK signaling in mature adipocytes-highlighting the context-dependent nature of GPR146-mediated metabolic control. In the absence of GPR146, impaired adipocyte differentiation and reduced lipolytic capacity together lead to diminished efflux of free fatty acids (FFAs) from adipose tissue. This reduction in FFA release limits hepatic FFA influx, thereby contributing to the observed protection against hepatic triglyceride accumulation and diet-induced fatty liver. These findings therefore position GPR146 as a key regulator of adipose-liver metabolic crosstalk under conditions of metabolic stress.

## Discussion

Human genetic studies have been instrumental in identifying novel regulators of metabolic disease, with GPR146 initially uncovered as a cholesterol-associated locus through genome-wide association studies (GWAS). Our previous work established that GPR146 regulates plasma LDL-C levels by modulating hepatic cholesterol biosynthesis and VLDL secretion[12]. Notably, the same SNPs in the locus are strongly associated with circulating liver enzyme levels in humans[17], suggesting broader involvement in hepatic and systemic metabolism. In this study, we reveal a previously unrecognized role for GPR146 in the regulation of hepatic steatosis and obesity development. Using both constitutive and acute depletion mouse models, we show that GPR146 deficiency confers resistance to diet-induced obesity and hepatic steatosis, accompanied by reduced circulating FFAs. These protective effects are primarily mediated through adipose tissue rather than the liver, highlighting the importance of adipose–liver crosstalk in the pathogenesis of MASLD. Mechanistically, we demonstrated that GPR146 promotes adipogenesis in preadipocyte via Gαq–PKC–AKT signaling and enhance lipolytic activity in mature adipocytes through ERK activation. Together, our findings demonstrated GPR146 as a key regulator of inter-organ metabolic communication and a potential therapeutic target for obesity and MASLD.

Prompted by GWAS associations with liver enzymes, we first evaluated hepatic lipid metabolism. Both constitutive and acute GPR146 depletion resulted in reduced hepatic triglyceride content and lower expression of genes encoding pro-inflammatory cytokines indicating protection against hepatic steatosis and related inflammatory stress. Liver metabolomic profiling in whole body knockout mice revealed elevated levels of PPP intermediates, suggesting increased NADPH generation. NADPH is known to support redox homeostasis by fueling glutathione and thioredoxin systems, thereby mitigate oxidative stress, which is a key driver of liver inflammation. Although NADPH was not directly measured, this inferred increase is consistent with the observed downregulation of inflammatory gene expression in the liver and may reflect improved redox buffering capacity. In parallel, increased hepatic glycogen content further support redirection of glucose toward storage rather than lipogenesis. The co-occurrence of glycogen accumulation and PPP metabolite enrichment may reflect altered glucose partitioning under insulin responsive conditions, together indicating improved hepatic insulin action.

Notably, increased respiratory exchange ratio (RER) under chow-fed conditions supports a glucose-favoring systemic profile, aligning with improved hepatic glucose handling. This shift was accompanied by increased lean mass in chow-fed male knockouts, suggesting a broader adaptation in body composition that may contribute to enhanced glucose utilization under non-stressed conditions. The absence of RER elevation under high-fat diet conditions, where dietary lipids dominate, underscores the context-dependent manifestation of this metabolic flexibility. Together, these observations point to the role of GPR146 in orchestrating hepatic and systemic metabolic adaptation that enhance glucose handling and support metabolic health across nutritional states.

Building upon the observed shifts in hepatic glucose handling and systemic substrate utilization, we next examined how GPR146 deficiency influences energy expenditure, revealing striking sex-specific differences. In female whole-body knockout mice on HFD, energy expenditure was markedly increased during dark phase and was accompanied by strong induction of *Ucp1* expression in eWAT. This suggests that female adipose tissue may exhibit greater thermogenic adaptability when GPR146 is absent, a response likely facilitated by the female hormonal milieu.

In contrast, male knockout mice exhibited a modest increase in energy expenditure limited to the light phase, without transcriptional activation of *Ucp1* or its upstream regulators. While canonical UCP1-mediated thermogenesis does not appear to be engaged at the mRNA level, we cannot exclude the possibility of post-translational modulation of UCP1 activity[42,43]. Though transcriptomic signatures suggest potential involvement of ATP-consuming futile lipid cycling, this remains speculative. Notably, Seahorse assays in GPR146-deficient adipocytes showed no increase in oxygen consumption, suggesting that the elevated energy expenditure observed in vivo likely reflects systemic or chronic adaptations not recapitulated in vitro (Supplementary Fig. 6j). Further studies will be necessary to elucidate the underlying mechanism in males.

In light of the tissue-specific knockout findings implicating adipose GPR146 in the systemic phenotype, we next considered its role in adipocyte biology. Rather than acting as a binary switch, GPR146 appears to modulate adipocyte differentiation in a dose- and stage-dependent manner. Unlike master regulators such as PPARγ, whose deletion leads to lipodystrophy and severe metabolic dysfunction, loss of GPR146 attenuates lipid accumulation without impairing early adipocyte commitment, resulting in smaller but histologically intact adipose depots. This interpretation is supported by the selective downregulation of mid-to-late stage adipogenic markers such as *Pparg* and *Cebpa*, while early regulators like *Cebpb* remain largely unaffected. Furthermore, both knockdown and overexpression of GPR146 in human preadipocytes impair differentiation, underscoring the requirement for tightly regulated GPR146 expression to support effective adipogenic progression.

Interestingly, transcriptomic meta-analysis from the Adipose Tissue Knowledge Portal revealed that GPR146 expression is significantly lower in subcutaneous adipose tissue of individuals with obesity compared to controls without obesity (Supplementary Fig. 8f). This inverse relationship contrasts with the phenotypes observed in GPR146-deficient mice but may reflect compensatory downregulation in response to metabolic stress or shifts in adipose tissue cellular composition during obesity[33]. Given that GPR146 is most highly expressed in mature adipocytes, obesity-associated changes such as increased immune infiltration and stromal remodeling may reduce the relative abundance of GPR146-expressing cells in bulk tissue profiles. Together, these findings highlight the complex regulation of GPR146 in adipose tissue and suggest that its functional role may not be fully captured by steady-state expression levels alone.

Taken together, our comprehensive investigation identifies GPR146 as a key regulator of adipose–liver metabolic coordination. GPR146 deficiency confers a distinct metabolic profile characterized by protection against diet-induced obesity and hepatic steatosis, along with reduced circulating free fatty acid levels and suppression of inflammatory gene expression. These benefits were observed across whole-body, adipose-specific, and AAV-mediated depletion models. The observed sex-specific responses underscore the importance of incorporating biological sex into future therapeutic development. Together, these findings nominate GPR146 as a promising therapeutic target for metabolic dysfunction–associated steatotic liver disease.

## Methods

### Mice and diet

All animal care and experimental procedures conducted in this study received approval from the Institutional Animal Care and Use Committee of Harvard University and National University of Singapore. The mice were maintained in a controlled environment with a temperature of $23 \pm 2\,°C$ and approximately 40% humidity, following a 12-h light-dark cycle with free access to water and food unless otherwise stated. Male and female mice aged between 6 to 36-week-old were utilized in this study to ensure matching of age and sex. C57BL/6 mice were acquired from the Jackson Laboratory at 8 weeks of age. $Gpr146^{+/+}$ and $Gpr146^{-/-}$ mice were generated through breeding of $Gpr146^{+/-}$ mice. Adipose-specific and liver-specific GPR146 knockout mice were established based on our previous study by crossing $Gpr146^{fl/fl}$ mice with $Adipoq$-Cre mice or $Albumin$-Cre mice obtained from the Jackson Laboratory[12]. Additionally, Rosa26-Cas9 knockin mice were also purchased from the Jackson laboratory as well. For dietary intervention studies, mice were subjected to a high-fat diet (60/Fat, TD.06414, Envigo) for indicated period of time.

For AAV-mediated knockdown studies, 8-week-old male C57BL/6 mice received retro-orbital injections of adeno-associated virus serotype 8 (AAV8) carrying either a scramble control or Gpr146-targeting shRNA under the U6 promoter. One week post-injection, mice were placed on a MASH-inducing diet enriched in fat, cholesterol and fructose (TD160785, Envigo) for 13 weeks. At study endpoints, mice were euthanized by carbon dioxide ($CO_2$) inhalation followed by cervical dislocation, in accordance with institutional animal care and use guidelines.

### Indirect calorimetry and metabolic studies

Body composition was assessed using an EchoMRI-100 T system (Echo Medical Systems) in multiple cohorts of Gpr146-deficient mice, including whole-body knockout ($Gpr146^{-/-}$) and control ($Gpr146^{+/+}$) mice (both sexes) fed either chow or high-fat diet (HFD), as well as adipose-specific knockout mice subjected to HFD feeding. Indirect calorimetry and locomotor activity were continuously monitored using a comprehensive lab animal monitoring system (CLAMS; Columbus Instruments). Measurements were conducted after 2 weeks of HFD feeding in whole-body knockout mice and at later time points in tissue-specific knockout models. Energy expenditure was calculated based on oxygen consumption ($VO_2$), and body weights were recorded before and after the metabolic cage study.

Fasting glucose and fasting insulin levels were measured in an independent cohort that included both male and female $Gpr146^{+/+}$ and $Gpr146^{-/-}$ mice maintained on either standard chow or HFD for 2 months. Blood glucose was determined using a handheld glucometer (OneTouch Ultra Easy, LifeScan) following an overnight fast, and plasma insulin levels were quantified by ELISA according to the manufacturer's protocol (Abcam).

Plasma free fatty acid (FFA) concentrations were measured using the Free Fatty Acid Quantification Kit (Sigma-Aldrich) in separate cohorts of mice, including both sexes of whole-body $Gpr146^{-/-}$ and $Gpr146^{+/+}$ mice on HFD, as well as adipose-specific and liver-specific knockout models following HFD feeding. All assays were performed in accordance with the manufacturers' instructions.

### Glucose tolerance tests (GTTs) and insulin tolerance tests (ITTs)

For glucose tolerance tests, $Gpr146^{+/+}$ and $Gpr146^{-/-}$ mice of both genders, which had been fed with chow or HFD for 2 months were individually housed and subjected to a 16 h-fasting before the assay. After baseline blood glucose levels were measured using One Touch Ultra (LifeScan), glucose (1 g/kg body weight) was injected through intraperitoneal route. Glucose levels were further determined at 15, 30, 60, 90, and 120 min post-injection. The insulin tolerance test was conducted after 4-h fasting of the mice. After baseline blood glucose levels were measured, human insulin (0.75-1U/kg body weight) was injected, followed by monitoring the blood glucose levels at 15, 30, 60, 90, and 120 min post-injection.

### Hepatic lipid content assay

Hepatic lipid extraction was perform as previously described with slight modifications[44]. In brief, snap-frozen liver samples were weighed and homogenized in ten volumes of ice-cold PBS. Thereafter, 200 μl of the homogenate was transferred into 1200 μl of chloroform: methanol (2:1; v/v) mixture, followed by vigorous vortex for 30 s. One-hundred microliters of ice-cold PBS was then added into the mixture and mixed vigorously for 15 s. The mixture was subsequently centrifuged at 1680 x g for 10 min at 4 °C. Two-hundred microliters of the organic phase (bottom layer) was transferred into a new tube and evaporated for dryness. To dissolve the dried lipids, 200 μL of 1% Triton X-100 in ethanol was added, and the solution was continuously rotated for 2 h. Triglyceride, cholesterol, and free fatty acid content were determined using the Infinity Triglycerides reagent (Thermo Scientific), Infinity Cholesterol reagent (Thermo Scientific), and the free fatty acid quantification assay kit (Abcam), respectively, following the manufacturer's instructions.

### Isolation of the mouse SVF

Subcutaneous adipose tissue depots were dissected from male wildtype C57BL/6 J or Rosa26-Cas9 knockin mice aged 8–11 weeks. The tissue was finely minced using scissors and then digested in KRPH buffer (20 mM HEPES,5 mM KH2PO4,1 mM MgSO4.7H2),1mMCaCl2.2H2O,136 mM NaCl,4.7 mM KCL supplemented with 1 mg/ml collagenase A (Sigma-Aldrich) for 1 h at 37 °C with gentle agitation. Following digestion, the tissue suspension was filtered through a 100 μm cell strainer and then centrifuged at 150 × g for 10 min at room temperature. The floating adipocytes fraction was subsequently collected, while the pellet was resuspended in DMEM/F12 medium supplemented with 10% fetal bovine serum (FBS) and 1% penicillin–streptavidin (P/S) (Thermo Fisher Scientific) as stromal vascular fraction (SVF). The medium was replaced the following morning to remove any floating cells or debris.

### Cell lines, culture conditions, transfection and infection

The Human Simpson–Golabi–Behmel syndrome (SGBS) cell line was kindly supplied by Martin Wabitsch (University of Ulm, Germany)[45] and authenticated as preadipocytes based on their response to adipogenic cues (cocktails) and the expression of adipogenic marker genes. The SGBS cell line was originally derived from human adipose tissue obtained with informed consent and with approval from the Ethical Committee of the University of Ulm, as described previously[45]. No additional ethical approval was required for its use in the present study. The SGBS cells were cultured in a basal medium of DMEM/F12 medium supplemented with 10% FBS, 3.3 mM biotin,1.7 mM pantothenic acid, and 100 U/ml penicillin-streptomycin (Sigma-Aldrich). To knock down the expression of $GPR146$ in human SGBS or mouse SVF cells, RNAimax (Thermo Fisher Scientific) reagent was used following the manufacturer's instructions. To knock out $Gpr146$ in mouse SVF cells isolated from Rosa26-Cas9 mice, lentivirus expressing sgRNAs targeting GPR146 were incubated with SVF cells for 16 h, after which the medium was replaced. For overexpression studies, SGBS cells were transduced with doxycycline-inducible lentiviral vectors expressing GFP or $GPR146$ and treated with 1 μg/mL doxycycline during the experimental period as indicated.

### Adipocyte differentiation

Adipogenic differentiation of SGBS cell was induced by treating confluent cells with serum-free quick-Diff medium comprising 0.01 mg/ml transferrin, 20 nM insulin, 100 nM cortisol, 0.2 nM tri-iodothyronine (T3), 25 nM dexamethasone, 250 μM isobutyl-1-methylxanthine (IBMX), and 2 μM rosiglitazone (Sigma-Aldrich). After 4 days, the

medium was replaced with DMEM/F12 containing 3.3 mM biotin,1.7 mM pantothenic acid, 0.01 mg/ml transferrin, 20 nM insulin, 100 nM cortisol, 0.2 nM tri-iodothyronine every other day (Sigma-Aldrich).

For differentiation of mouse SVF cells, the cells were first cultured to confluence in DMEM/F12 medium supplemented with 10% FBS and 1% P/S, induced to differentiation in the adipogenic cocktail consisting 1 μM dexamethasone, 5 μg/ml Insulin, 0.25 mM IBMX, 5 μM rosiglitazone, followed by maintenance treatment with 1 μg/ml Insulin and 5 μM Rosiglitazone every 48 h until fully differentiated. Eight to ten days after differentiation, the differentiated adipocytes were stained with BODIPY (Sigma-Aldrich) or collected for RNA and protein extraction.

### Quantification of in vitro differentiation

**Oil-red O staining.** For Oil-red O staining, differentiated adipocytes were rinsed with PBS twice and then fixed in 4% (v/v) paraformaldehyde solution for 20 min. Thereafter, the fixed cells were washed twice with PBS, followed by a 5 min incubation with 60 % isopropanol. The cells were then stained with Oil-Red O isopropanol solution for 15 min at room temperature, washed with PBS and imaged.

**BODIPY staining.** To assess differentiation potential, fully differentiated adipocytes were stained with 1 μM BODIPY for 20 min, followed by washing with PBS and imaged using the PerkinElmer Operetta high-content imaging system. The percentage of BODIPY-positive cells and the areas of the BODIPY stained adipocyte were quantified using Image J.

**Histological analyses.** Following dissection, epididymal white adipose tissue (eWAT), inguinal white adipose tissue (iWAT) and liver were promptly fixed in a 10% formalin-buffered solution (Sigma-Aldrich). Next, the fixed tissues were embedded, sectioned, and stained with haematoxylin and eosin (H&E) (Sigma-Aldrich). Histopathological images were captured with a digital pathology scanner (Aperio Versa 200, LEICA) and analysed with Image J (National Institutes of Health) and Adiposoft[46] software to measure the area of adipocytes. The areas of 150–300 cells from two sections from each sample were measured.

**In vitro and in vivo lipolysis.** To assess the role of GPR146 in lipolysis, both in vitro and in vivo assays were performed. For in vitro analysis, differentiated human SGBS adipocytes were subjected to *GPR146* knockdown using siRNA (ThermoFisher) or overexpression via doxycycline-inducible lenti-viral vectors expressing either GPP (control) or *GPR146*. Glycerol release was measured under basal and forskolin-stimulated conditions using a colorimetric assay kit (Sigma-Aldrich).

For in vivo lipolysis, male mice with AAV-mediated *Gpr146* knockdown or scramble control, following 6 weeks of MASH diet feeding, were fasted for 4 h and injected intraperitoneally with β3-adrenergic agonist CL-316,243 (1 mg/kg body weight) (MCE). Plasma glycerol levels were quantified at baseline, 30-, 60-, and 120-min post-injection.

**cAMP/Gs and Gαq- IP1assay.** To investigate the signaling downstream of GPR146, cellular cAMP level, which reflects the activation status of Gαs or Gαi signaling, was quantified using cAMP-Gs dynamic kit (PerkinElmer) following the manufacture's protocol. Human SGBS cells or mouse SVF cells were transfected with siRNA to knock down the expression of *GPR146* and then induced to differentiation with adipogenic cocktail. Cells were collected on day 0,1,2,3, and 4 after differentiation, resuspended in PBS and used for the cAMP-Gs assay. Specifically, 7 μl of the cell suspension was incubated with HTRF reagents (kit components) for 1 h at room temperature, followed by the measurement of the fluorescence ratio at 665/620 nm. Data were

reported as the mean ± standard error of the mean (SEM) of three replicates.

Similarly, the activation status of Gαq signaling was probed following *GPR146* knockdown by performing an IP1 assay using the IP-One Gq Kit (PerkinElmer). Transfected human SGBS preadipocytes and mouse SVF undergoing differentiation were harvested at indicated time points, and the endogenous IP1 level was quantified by measuring HTRF signal at 665/620 nm.

**Seahorse XF mitochondrial stress test.** SGBS preadipocytes were differentiated into mature adipocytes as previously described. Fully differentiated cells were transfected with control or GPR146 siRNA, and 48 h after transfection, mitochondrial respiration was assessed using the Seahorse XF Cell Mito Stress Test Kit (Agilent Technologies) according to the manufacturer's instructions. Oxygen consumption rate (OCR) was measured using a Seahorse XFe24 Analyzer (Agilent Technologies) under basal conditions and following sequential injection of oligomycin, FCCP, and rotenone/antimycin A.

**Metabolomics.** Mouse liver samples (average weight of 53.5 mg) were homogenized in four volumes of water per milligram of tissue (4 μL/mg) using a TissueLyser II bead mill (QIAGEN), with two 3 mm tungsten carbide beads and two 120-second intervals of high-speed shaking at 20 Hz. Homogenates were then aliquoted for analyses of lipids and polar metabolites using four complementary liquid chromatography tandem mass spectrometry (LC-MS) procedures as described previously[47].

**HILIC–positive mode (HILIC-pos).** Water-soluble metabolites were analyzed using a Shimadzu Nexera X2 UHPLC coupled to a Q Exactive Hybrid Quadrupole Orbitrap mass spectrometer (Thermo Fisher Scientific). Liver homogenates (10 μL) were protein-precipitated with nine volumes of acetonitrile/methanol/formic acid (74.9:24.9:0.2, v/v/v) containing stable isotope-labeled internal standards (valine-d8 and phenylalanine-d8). After centrifugation ($9000 \times g$, 10 min, 4 °C), supernatants were injected onto an Atlantis HILIC column (150 × 2 mm, 3 μm; Waters). Elution was performed at 250 μL/min starting with an isocratic hold at 5% mobile phase A (10 mM ammonium formate, 0.1% formic acid) for 0.5 min, followed by a linear gradient to 40% mobile phase B (acetonitrile with 0.1% formic acid) over 10 min. MS data were acquired in positive electrospray mode (m/z 70–800) at 70,000 resolution. Source settings were: sheath gas 40, sweep gas 2, spray voltage 3.5 kV, capillary temperature 350 °C, heater temperature 300 °C, S-lens RF 40, microscans 1, AGC target 1e6, and maximum ion time 250 ms.

**HILIC–negative mode (HILIC-neg).** Targeted analysis of polar metabolites was performed on an ACQUITY UPLC system (Waters) coupled to a 5500 QTRAP mass spectrometer (SCIEX). Liver homogenate (30 μL) was extracted with four volumes of 80% methanol containing isotope-labeled internal standards (inosine-15N4, thymine-d4 and glycocholate-d4). Following centrifugation ($9000 \times g$, 10 min, 4 °C), supernatants were analyzed using a Luna $NH_2$ column (150 × 2.0 mm; Phenomenex) at a flow rate of 400 μL/min. The gradient began with 10% mobile phase A (20 mM ammonium acetate, 20 mM ammonium hydroxide in water) and 90% mobile phase B (10 mM ammonium hydroxide in 75:25 acetonitrile/methanol), ramping linearly to 100% A over 10 min. Detection was carried out in negative electrospray mode using multiple reaction monitoring, with compound-specific declustering potentials and collision energies optimized using authentic standards. The ion spray voltage was -4.5 kV and the source temperature was 500 °C.

**Lipid profiling (C8-positive).** For global lipid profiling, liver homogenates (10 μL) were mixed with 190 μL of isopropanol containing 1,2-

didodecanoyl-sn-glycero-3-phosphocholine (Avanti Pola Lipids; Alabaster, AL) as an internal standard to precipitate proteins and extract lipids. Following centrifugation, the supernatants were injected directly onto a ACQUITY BEH C8 column ($100 \times 2.1$ mm, 1.7 µm; Waters). The mobile phases were (A) 95:5:0.1 (v/v/v) 10 mM ammonium acetate/methanol/formic acid and (B) 99.9:0.1 (v/v) methanol/formic acid. The elution gradient began with an isocratic hold at 80% A for 1 min, followed by a linear ramp to 80% B over 2 min, a linear increase to 100% B over 7 min, and a final isocratic hold at 100% B for 3 min. MS data were acquired in positive electrospray ionization (ESI) mode using full-scan analysis over 200–1000 m/z at 70,000 resolution and 3 Hz data acquisition rate. Source parameters were set as follows: spray voltage 3 kV, capillary temperature 300 °C, heater temperature 300 °C, sheath gas 50, sweep gas 5, S-lens RF 60, in-source CID 5 eV, microscans 1, automatic gain control target 1e6, and maximum ion time 100 ms. Lipid species were identified by comparison with reference plasma extracts and annotated based on the total acyl chain carbon count and double bond content.

**Reversed-phase analysis (C18–negative).** Analyses of free fatty acids, bile acids, and metabolites of intermediate polarity were conducted using a Shimadzu Nexera X2 U-HPLC coupled to a Q Exactive mass spectrometer (Thermo Fisher Scientific). Liver homogenates (30 µL) were extracted using 90 µL of methanol containing PGE2-d4 as an internal standard (Cayman Chemical Co.; Ann Arbor, MI) and centrifuged (10 min, $9000 \times g$, 4 °C). The supernatants (10 µL) were separated on an ACQUITY BEH C18 column ($150 \times 2.1$ mm; Waters) at a flow rate of 450 µL/min. The mobile phases consisted of (A) 0.01% formic acid in water and (B) 0.01% acetic acid in acetonitrile. The gradient started isocratically with 20% A for 3 min, followed by a linear gradient to 100% mobile phase B over 12 min. MS detection was performed in negative ESI mode (m/z 70-850) at a resolution of 70,000 (3 Hz). Additional MS settings included: spray voltage, -3.5 kV; capillary temperature, 320 °C; probe heater temperature, 300 °C; sheath gas, 45; auxiliary gas, 10; and S-lens RF level 60.

**Data processing.** Raw data files from the Exactive Plus and Q Exactive instruments were processed using TraceFinder (Thermo Fisher Scientific) and Progenesis QI (Nonlinear Dynamics). Data acquired on the 5500 QTRAP were processed using MultiQuant (SCIEX). For each method, metabolite identities were confirmed using authentic reference standards or reference samples.

**Immunoblotting.** For immunoblotting analyses, cells or tissues were lysed in RIPA buffer containing the protease and phosphatase inhibitor cocktail, and the protein concentration was estimated using a BCA protein assay kit (Thermo Fisher Scientific). Equal amounts of protein were subjected to SDS-PAGE, transferred to polyvinylidene fluoride (PVDF) membranes, followed by incubations with primary and secondary antibodies.

**Gene expression analysis.** Total RNA was extracted from cells or tissues using Trizol (Thermo Fisher Scientific) or the RNeasy kit (QIAGEN). 1 µg of RNA was reverse transcribed using High Capacity cDNA Reverse transcription kit (Thermo Fisher Scientific). Quantitative real-time PCR(RT-PCR) was performed on a ViiATM 7 RT-PCR system (Applied Biosystems Inc.) with SYBR-green ChamQTM SYBR Color qPCR Master Mix (Vazyme). Relative expression level for gene of interest was compared to the housekeeping gene RPLP. Primer sequences used in this study was listed in supplementary table 2.

Genome-wide mRNA expression profiles in mouse eWAT, iWAT, BAT, and liver were assessed with the Affymetrix GeneChip™ Mouse Gene 2.1 ST Array Strip in accordance with the manufacturer's instructions. Gene set enrichment analysis (GSEA) was performed, and

significantly enriched terms were identified using a false discovery rate q value of less than 0.05.

**Phosphoproteomics**

**Sample preparation.** Three replicates of control and GPR146 knock down SGBS cells were harvested by scraping into lysis buffer (8 M urea, 50 mM Tris-HCl, pH 8.0 with protease and phosphatase inhibitor cocktails) directly after two PBS washes. The cell pellets were sonicated for 3 cycles and the protein concentration was determined using Pierce BCA Protein Assay Kit following the manufacturer's instructions. All samples (approximately 200 µg each) were subjected to reduction with 20 mM tris (2-carboxyethyl) phosphine (TCEP) in 100 mM triethylammonium bicarbonate buffer (TEAB) and alkylated by 55 mM chloroacetamide (CAA) at room temperature for 30 min in the dark. The urea concentration was then diluted to 1 M using 100 mM TEAB, pH 8.5 for digestion. The samples were digested with 5 µg of Lys C for 4 h, followed by overnight digestion with 5 µg of modified trypsin at 25 °C. The efficiency of digestion was evaluated and considered acceptable if protein missed cleavage was 10% or lower. The resulting peptides were acidified and desalted with a 60 mg HLB cartridge (Waters) as per manufacturer's instructions. Briefly, the cartridge was first activated with 0.5 ml of 100% acetonitrile and then equilibrated with 1 ml of 0.5% acetic acid in water, followed by peptide sample loading. Bound peptides were washed twice with 2 mL of 0.5% acetic acid in water and eluted with 0.5 mL of 0.5% acetic acid in 65% acetonitrile. Approximately one-tenth of the desalted samples was retained as a proteome input control, while the remaining samples were used for phosphorylated peptide enrichment.

Phosphopeptides were enriched from the desalted peptides with TiO₂ beads. The samples were resuspended in a final composition of 6% trifluoroacetic acid (TFA) in 40% acetonitrile and incubated with the TiO₂ beads at 25 °C for 10 min. This incubation step was repeated once more with fresh beads. Beads from both enrichments were combined, and a sequential wash was performed with 6% TFA in 10%, 40%, and 60% acetonitrile. The elusion was carried out first with 5% ammonia water first followed by 5% ammonia water in 25% acetonitrile.

Both the input and phosphopeptides samples were dried using vacuum centrifugation, resuspended with 100 mM TEAB (pH 8.5) and labeled individually for 2 h with TMTsixplex™ Isobaric Label Reagent (Thermo Fisher Scientific). After quenching with 1 M ammonium formate (pH 10), the labeled samples were pooled together according to their respective proteomics input set and phosphopeptide set. Subsequently, the samples were desalted and eluted with 10 mM ammonium formate (pH 10) in increasing acetonitrile concentration of 14%, 18%, 21%, 24%, 27%, 32% and 60%, resulting a total of 7 fractions for each sample set. Following two acidified washes, the fractions were ready for data acquisition.

**Mass spectrometry.** Both the proteomics input samples and phosphopeptides samples were subjected to mass spectrometry analysis using reverse phase liquid chromatography on the nLC1000 UHPLC system connected to Thermo Orbitrap Fusion Lumos mass spectrometer (Thermo Fisher Scientific). Each fraction was separated on a 50 cm length x 75 µm internal diameter, 2 µm particle size Easy Spray column (Thermo Fisher Scientific) over a 70-min gradient with pre-programmed mixing of solvent A (0.1% formic acid in water) and solvent B (99% acetonitrile, 0.1% formic acid in water). The following acquisition parameters were applied for all samples: data dependent acquisition (DDA) in positive ion mode with survey scan of 60,000; scan range of 350–1550 m/z, and AGC target of 4e5; MS/MS analysis using Orbitrap mass analyzer at higher energy collision dissociation of 42%; AGC target of 7.5e4; isolation window of 1 m/z; and maximum injection time of 100 ms.

**Quantification and statistical analyses.** The statistical significance of differences was determined by unpaired, two-tailed Student's t test (two groups) or oneway ANOVA (more than two groups), corrected p values < 0.05 considered statistically significant. Asterisks denote corresponding statistical significance *p < 0.05 and **p < 0.01. For mouse liver western blotting, each sample corresponds to protein extract from one mouse.

## Reporting summary

Further information on research design is available in the Nature Portfolio Reporting Summary linked to this article.

## Data availability

The microarray data generated in this study have been deposited in the Gene Expression Omnibus (GEO) database under accession number GSE316218. The phosphoproteomics and proteomics data have been deposited to the ProteomeXchange Consortium via the PRIDE partner repository with the dataset identifier PXD074087. Processed metabolomics and lipidomics data have been deposited in Figshare. Source data underlying the figures are provided with this paper. All other data supporting the findings of this study are available within the Article and Supplementary Information or from the corresponding author upon request. Source data are provided with this paper.

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

## Acknowledgements

This study was supported by the National University of Singapore Start-up grant (H.J.Y.) and Singapore MOE Academic Research Fund Tier 2 grant MOE-T2EP30221-0013 (H.J.Y.). We thank Dr. Ben Zhou, Dr. Lianfeng Wu for their invaluable support and discussions.

## Author contributions

H.Y. performed conceptualization, supervision, and funding acquisition. Y. Shi, H.Y. performed methodology. Y. Shi and H.Y. performed formal analysis. Y. Shi., K. Y. C, T. T. T., Y. W., Y. Y., X. C., V. C., Y. Shen., Y. H., T. Z., Y. T. L., A. D., C. D., K. P., K. B., M. W., C. B. C., A. S. B., R. M. S., C. C. and H. Y. performed investigation. Y. Shi and H.Y. wrote the original draft. H.Y. provided resources.

## Competing interests

C.A.C. is a founder of CRISPR Therapeutics, Sana Biotechnology, and CLADE Therapeutics. The remaining authors declare no competing interests.
