## [Transparent Peer Review file · Nature Communications]

GPR146 in adipose tissue drives adipose-liver crosstalk and promotes hepatic steatosis

Corresponding Author: Dr Haojie Yu

Version 0:

Reviewer comments:

Reviewer #1

(Remarks to the Author)

The authors present findings that reveal a role for GPR146 in the prevention of hepatic steatosis. They propose that it is GPR146's role in adipose tissue and lipid metabolism that prevents TG accumulation in the liver. GPCRs represent attractive drug targets and GPR146 has intriguing biology, however, several key questions remain.

If GPR146 KO affects adipogenesis, is that really desirable? The authors show similarities in pattern to the master adipose regulator Pparg and yet knocking out Pparg/disrupting adipogenesis is deleterious on metabolic health, including the liver. Especially since knocking down GPR146 affects hyperplasia (which is the healthier expansion compared to hypertrophy). If the adipose tissue isn't expanding and the mice eat the same amount of HFD, as shown by the authors, then where is the extra fat going? Finally, if GPR146 affects adipogenesis, why don't they observe lipodystrophy in the whole body or adipose specific KO?

Regarding the energy expenditure, several questions remain. Is the increase in catabolism that was observed in vivo also present in isolated adipocytes? What is the in vivo energy expenditure profile in the adipose specific KOs? Does the increased energy expenditure persist at thermoneutrality? How about on chow? The authors say the mechanism is not through canonical thermogenic cycles but Adrb3, which they show is up, is the major driver of both Ucp1 and non-Ucp1 dependent thermogenesis. And CL-316,243 does far more than just stimulate lipolysis.

Why do the author's think that knocking Gpr146 down in cells has an effect when they haven't added any ligand? Is it known if the ligand for GPR146 is endogenously produced?

All the in vivo findings involve prevention of steatosis because of constitutively expressed genetic models. Inducibly depleting Gpr146 after NAFLD has developed would significantly strengthen the manuscript.

Minor

-In the discussion, just because Ucp1 expression is unchanged does not mean the EE is Ucp1 independent.

Reviewer #2

(Remarks to the Author)

In this manuscript, the authors investigated the regulatory role of the G protein-coupled receptor 146 (GPR146) in adipose tissue and liver lipid metabolism. In an in vivo model, GPR146 deficiency exhibited a protective effect against obesity induced by a high-fat diet, reducing liver steatosis and accompanying a decrease in circulating liver enzyme levels. Global depletion of GPR146 in mice affected energy homeostasis by increasing energy expenditure during high-fat diet feeding. The study revealed that specific depletion of GPR146 in adipocytes in mice could counteract diet-induced fatty liver by limiting the influx of free fatty acids. Additionally, the authors demonstrated that GPR146 deletion in vitro inhibits adipogenesis by blocking the activation of $G\alpha_q$ -PKC AKT signaling.

The subject matter of this study holds considerable significance in the realm of lipid metabolism. Nevertheless, the data outlined in the current version do not entirely substantiate the authors' conclusions. A notable deficiency lies in the absence of a mechanistic explanation detailing how GPR146 influences lipid turnover and adipogenesis. Here are specific comments

aimed at potentially enhancing the manuscript.

1. The first question inquires whether the protein expression level of GPR146 is elevated in the adipose tissue of obese individuals compared to non-obese ones.
2. Does the expression level of GPR146 in wild-type mice also increase over time in various adipose tissues (eWAT, ingWAT, and BAT) during HFD feeding?
3. Global depletion of GPR146 in mice, as observed in Fig 2b by MRI measurements, led to an increase in the percentage of lean mass under HFD feeding. However, the data only indicate the percentage of lean mass. Is there an increase in the absolute amount of lean mass as well? Utilizing a whole-body composition analyzer might offer more precise *in vivo* measurements the mass of body lean, fat, and fluid.
4. Does the protective role of GPR146 deficiency against high-fat diet (HFD)-induced obesity suggest a contrasting effect when GPR146 is up-regulated? The authors should provide or at least discuss the results induced by GPR146 over-expressions.
5. Moreover, the data from GPR146 deficiency do not entirely support the authors' conclusions regarding the promoting effects of GPR146 on adipogenesis. To strengthen this assertion, the author should include data from GPR146 overexpression experiments.
6. The author discussed ATP-consuming futile cycles in whole-body GPR146 deficiency. It raises the question whether similar findings could be observed in mice with adipocyte- or hepatocyte-specific GPR146 deficiency.
7. Since PKC is the endpoint of the proposed GPR146 signaling pathway, in the GPR146 depletion background, inhibition of PKC function by pharmacological reagent alone cannot support the authors' hypothesis that $G\alpha_q$ -PKC-pAkt is downstream of GPR146 signaling.
8. To address the roles of GPR146 in lipid turnover, the authors should provide more functional experiments, instead of just differential expressions analyses.

Minor points:

1. It would be helpful to provide a brief description of G protein-coupled receptors in mediating functional activities of adipose tissues.
2. The image should include a scale bar for reference.
3. Please include the sequences for the siRNA used.

Reviewer #3

(Remarks to the Author)

The study by Shi et al. provides a large set of data investigating the effects of GPR146 on liver steatosis and particularly the role of adipocyte GPR146. The authors first describe findings extracted from public GWAS databases indicating association of GPR146 variants with increased plasma liver enzymes and CRP. In the following experiments they show that GPR146 knockout reduces liver steatosis and plasma ALT in mice. Liver metabolomics was used to characterize in more detail effects on lipids. GPR146 knockout also decreased weight gain and fat mass in response to HFD that appears to be related to elevated energy expenditure during resting periods. The authors used tissue specific knockout models to demonstrate that Adipoq-Cre but not Alb-Cre induced knockout alleviated the liver steatosis suggesting that adipocyte GPR146 mediates the steatotic effect. Adipocyte KO of GPR146 also reduced weight gain, fat mass and plasma FFA. In cell models GPR146 was found to promote adipogenesis potentially through $G\alpha_q$ -PCK-AKT pathway. The study is interesting and thorough. However, there are several points of concern:

1. The effect of Adipoq-Cre induced knockout on Gpr146 expression in WAT requires further information. At least after 4 months of HFD the effect of the KO on Gpr146 expression appears either limited (ingWAT) or not existing (eWAT) (Extended Figure 7e). Please provide necessary evidence of the effect of Gpr146 KO on GPR146 expression in different time points used in the study. The current data raises questions about the mechanisms of the observed effects in Adipoq-KO mice.
2. The authors state in several instances that GPR146 KO ameliorates hepatic inflammation. However, I don't think any evidence is presented to support this notion. ALT cannot be considered an inflammation marker. However, indeed it would be necessary to study the role of GPR146 in relation to liver inflammation and fibrosis. Simple HFD model may not be the best for this purpose, but it may require additional factors such as fructose feeding. The authors also reported elevated CRP associated with GPR146 variants in humans. What about in mice? Does the Gpr146 KO affect CRP?
3. The effect on glucose metabolism should be better explained. Do the increased liver hexose metabolites indicate upregulation of hepatic glucose uptake or what is the authors' interpretation? It is not clear what the authors mean by statement that "GPR146 deficiency promotes glucose mobilization", page 5 line 9-10. Indeed, glycogen storage was increased.
4. The authors claim that GPR146 KO ameliorates insulin resistance. However, this is not obvious. The KO improves glucose tolerance (fig. 2g) which could suggest better insulin tolerance. However, in ITT the effects can be seen only in late

time points when the animals are recovering from the insulin effect. A true difference in insulin sensitivity should be seen in early time points when glucose is decreasing. This is especially obvious in female mice where the effect in ITT is observed only after 90 min (Ex. Fig. 3d) but also to some extent in male mice (Fig 2h). The results rather appear to suggest defects in counter-regulatory mechanisms correcting insulin-induced hypoglycemia. Also, there was no difference in plasma fasting insulin (Ex. Fig. 3e) again suggesting no clear effect on insulin sensitivity. HOMA-IR index, preferably calculated with the mouse modified formula (doi: 10.1177/0023677212473714) could be provided also.

5. The authors provide transcriptomics data suggesting decreased macrophage infiltration in the adipose deposits of the GRP146 KO mice. Is this supported by any other data such as histology?

6. The study includes several types of omics data; however, the full data has not been made available and no repository for data storage has been indicated?

Version 1:

Reviewer comments:

Reviewer #1

(Remarks to the Author)

I want to commend the authors are giving a strong rigorous effort to addressing my concerns and questions. Overall, I am quite satisfied. A few points remain for clarification:

-What is happening to the RER at the end of the Supp Figure 7 panel g?

-Quantification of the western blots would be beneficial

-Figure 3g-i has the EE for the females corrected by bodyweight. This should be changed to the data without bodyweight correction for accurate representation like the male data in the first part of Figure 3

(<https://pubmed.ncbi.nlm.nih.gov/40993210/>)

-Following up on the EE, I suppose the adipose KO mice in indirect calorimetry, paired with the cell data on isolated adipocytes, would suggest the EE is not arising from the fat tissue?

Reviewer #2

(Remarks to the Author)

In this revised version of the manuscript, the authors have made considerable efforts to strengthen the study by incorporating a substantial amount of new experimental data. The work now presents a more comprehensive and coherent narrative, encompassing large-scale data analyses, mouse experiments, and mechanistic investigations. Although some results do not fully align with the initial expectations, the authors have provided reasonable explanations to address these discrepancies. I have no further questions or concerns regarding this version of the manuscript.

Reviewer #3

(Remarks to the Author)

The authors have well addressed my concerns. No further comments.

Version 2:

Reviewer comments:

Reviewer #1

(Remarks to the Author)

I appreciate the authors rigorous efforts in addressing my comments, I have no further concerns.

Point-by-Point Response

We sincerely thank the reviewers for all insightful comments. In response, we have thoroughly revised the manuscript and addressed all reviewer comments with substantial new data and clarifications.

We sincerely apologize for the substantial delay in submitting this revision and greatly appreciate the patience and understanding of the reviewers and editorial team. After my transition to an independent faculty position, we encountered an unexpected issue: the original *Gpr146* whole-body knockout mouse strains had incorrect genotypes due to colony maintenance by a collaborator after the closure of my former postdoctoral lab. We therefore reconstructed and revalidated this line, which required considerable time and effort but was essential to ensure experimental rigor and reproducibility.

During this period, we have substantially expanded our investigation and uncovered new mechanistic insights, including the distinct roles of GPR146 in regulating adipogenesis and lipolysis via the Gαq-PKC-AKT and ERK signaling pathways, respectively. These findings further clarify how GPR146 modulates adipose-liver crosstalk and contributes to hepatic steatosis. We have also included new *in vivo* experiments using AAV-mediated knockdown with NASH diets to demonstrate the translational relevance of targeting GPR146 in adult animals.

In parallel, **we have taken great care to thoroughly restructure the manuscript to enhance clarity, logical flow, and scientific rigor.** Throughout, we have also tempered our language to avoid overinterpretation and ensure our conclusions are appropriately supported by the data. We hope this revised version fully addresses the concerns raised and is now suitable for further consideration.

REVIEWER COMMENTS

Reviewer #1 (Remarks to the Author):

The authors present findings that reveal a role for GPR146 in the prevention of hepatic steatosis. They propose that it is GPR146's role in adipose tissue and lipid metabolism that has intriguing biology, however, several key questions remain.

1. If GPR146 KO affects adipogenesis, is that really desirable? The authors show similarities in pattern to the master adipose regulator *Pparg* and yet knocking out *Pparg*/disrupting adipogenesis is deleterious on metabolic health, including the liver. Especially since knocking down GPR146 affects hyperplasia (which is the healthier expansion compared to hypertrophy). If the adipose tissue isn't expanding and the mice eat the same amount of HFD, as shown by the authors, then where is the extra fat going? Finally, if GPR146 affects adipogenesis, why don't they observe lipodystrophy in the whole body or adipose specific KO?

Answer: We thank the reviewer for this insightful and important question. We fully agree that excessive inhibition of adipogenesis can be deleterious to metabolic health, as highlighted in studies of *PPARG* deficiency. However, our data suggest that GPR146 does not serve as a binary on/off switch for adipogenesis but rather functions as a modulator of this process. And importantly, the reduced lipid storage capacity observed in GPR146-deficient mice is accompanied by increased energy expenditure, which likely represents the primary driver of the overall lean phenotype and diminished hepatic lipid accumulation.

Specifically, GPR146 deficiency does not abolish adipogenesis but leads to a partial attenuation of terminal differentiation, as evidenced by a selective reduction in mid-to-late

stage transcription factors (PPAR γ and C/EBP α), while early commitment factors such as C/EBP β remain unchanged (Fig. 6h, l). This suggests that GPR146 regulates adipocyte maturation rather than commitment. This allows for the formation of histologically intact, but smaller adipose depots, and importantly, we do not observe features of lipodystrophy in either the global (Fig. 2c, d, g, h) or adipose-specific knockout mice (Fig. 4k, l).

As GPR146 is most highly expressed in mature adipocytes, we further investigated its role in regulating lipolysis. In this revision, we present new data showing that doxycycline-induced overexpression of *GPR146* in SGBS-derived adipocytes significantly enhances lipolysis (new data, Fig. 8c) while both siRNA-mediated knockdown of *GPR146* in SGBS adipocytes (new data, Fig. 8b) and acute depletion of GPR146 in mice using AAV-shRNA significantly reduces lipolytic activity (new data, Fig. 8e). These findings are consistent with the reduction in circulating free fatty acid (FFA) levels observed in both global (Fig. 1k) and adipose-specific GPR146 knockout mice (Fig. 4e, f), which likely contribute to the reduced FFA influx into the liver and subsequent attenuation of hepatic TG accumulation.

At the whole-body level, in terms of energy balance, GPR146-deficient mice exhibit a significant increase in resting energy expenditure, particularly during the light (inactive) photoperiod in males (Fig. 3d). Honestly, while we present robust data demonstrating that adipose-specific depletion of GPR146 attenuates both adipogenesis and lipolysis—thereby reducing FFA efflux and contributing to protection against hepatic steatosis—the precise molecular mechanism by which GPR146 deficiency enhances energy expenditure remains unclear in male mice. Although we initially speculated the involvement of ATP-consuming futile cycles, our evidence is limited to mRNA expression data. Nevertheless, this elevated energy expenditure likely plays a key role in offsetting the reduced capacity for adipose expansion, thereby preventing ectopic lipid accumulation.

In our revised study, we also performed indirect calorimetry in female GPR146-deficient mice and found that KO females exhibited significantly increased energy expenditure compared to controls during both light and dark cycles (new data, Fig. 3h, i). Interestingly, in contrast to male KO mice where *Ucp1* expression remained unchanged in eWAT and ingWAT (Supplementary Fig. 6f, g), female KO mice displayed a more than 15-fold increase in *Ucp1* expression specifically in eWAT, along with marked upregulation of *Pgc1 α* and *Ppara* (new data, Supplementary Fig. 6h, i). These data suggest that in females, activation of UCP1-mediated thermogenesis likely contributes to the observed increase in energy expenditure, potentially through enhanced mitochondrial biogenesis and oxidative metabolism. This emerging sex difference in thermogenic activation may reflect differential hormonal regulation or adipose tissue plasticity between males and females^{1,2}, and further investigation to elucidate sex-specific mechanisms underlying GPR146-dependent metabolic control is required.

Importantly, we would like to highlight the systemic metabolic reprogramming in GPR146-deficient mice. The knockout mice exhibit improved glucose handling, accompanied by increased hepatic glycogen content and elevated levels of hexose and pentose phosphate intermediates, suggesting a possible enhancement in hepatic insulin sensitivity (Fig. 2k, l and Supplementary data Fig. 3a, b, c). The accumulation of pentose phosphate intermediates further suggests increased NADPH production³, which could contribute to redox homeostasis and protection from oxidative stress, a key driver of hepatic inflammation in NAFLD⁴.

Additionally, untargeted metabolomics also revealed significantly elevated hepatic levels of seven standard amino acids in *Gpr146*^{-/-} mice, namely histidine, cysteine, lysine, valine, leucine, isoleucine, and phenylalanine (new data, Supplementary Fig. 3d). Consistent with these hepatic metabolic changes, chow-fed *Gpr146*^{-/-} mice exhibited a significantly higher

respiratory exchange ratio (RER) during the dark cycle compared to controls (new data, Supplementary Fig. 3e, f), indicating a greater reliance on glucose as an energy source. Furthermore, chow-fed *Gpr146*^{-/-} mice displayed significantly increased lean mass without changes in fat mass (new data, Supplementary Fig. 3g, h), potentially reflecting altered systemic nutrient partitioning and increased availability of amino acids at chow-fed baseline condition. Together, these findings support a model in which GPR146 deficiency induces coordinated metabolic shifts that reduce the need for lipid storage and limit ectopic lipid accumulation in the liver.

We have revised the manuscript to reflect this updated mechanistic interpretation and thank the reviewer for prompting a deeper and more critical evaluation of these data.

Fig.8

Fig.3

Supplementary Fig.6

Supplementary Fig.3

Figure 8. GPR146 regulates mature adipocyte lipolysis through ERK signaling. **b**, Glycerol release from SGBS mature adipocytes transfected with siGPR146 or control siRNA, with or without forskolin stimulation; values normalized to intracellular triglyceride (n=3 replicates per experiment, representative of 3 independent experiments, by Student's t test). **c**, Glycerol release from SGBS adipocytes with doxycycline-inducible GPR146. Measurements were performed with or without the MEK inhibitor PD98059 and normalized to intracellular TG (n=3 replicates per experiment, representative of 3 independent experiments, by Student's t test). **e**, In vivo lipolysis assay in mice with acute depletion of *Gpr146* using AAV-shRNA or scramble control, maintained on a MASH-inducing diet for 12 weeks. Mice were injected with CL-316,243 to stimulate lipolysis, and plasma glycerol levels were

measured at the indicated time points. (n=7-9 mice per group, by Student's t test). * $p < 0.05$, ** $p < 0.01$; bars indicate mean \pm s.e.m..

Figure 3. GPR146-deficient Mice exhibit Elevated Energy expenditure. Indirect calorimetry study in female mice fed HFD for two weeks. **h,i**, Total energy expenditure (EE) during the light (**h**) and dark (**i**) phases, averaged over four consecutive days, expressed as kcal/kg lean mass. * $p < 0.05$, ** $p < 0.01$; bars indicate mean \pm s.e.m..

Supplementary Figure 6, Metabolic phenotyping and thermogenic pathway regulation in adipose tissue of GPR146-deficient and control mice, Related to Fig3. **h,i**, qPCR expression analysis of *Ucp1*, *Pgc1a* and *Ppara* in eWAT (**h**) and iWAT (**i**) from female *Gpr146*^{+/+} and *Gpr146*^{-/-} littermates fed HFD (n=4-8 mice per group, by Student's t test). * $p < 0.05$, ** $p < 0.01$; bars indicate mean \pm s.e.m..

Supplementary Figure 3. GPR146 deficiency reprograms hepatic glucose and amino acid metabolism. **d**, Relative abundance of hepatic amino acids in 16 h fasted male mice fed an HFD (n=7 mice per group, by Student's t test). **e,f**, Respiratory exchange ratio (RER) measured during the **dark** (**e**) and **light** (**f**) phases by indirect calorimetry in chow-fed *Gpr146*^{+/+} and *Gpr146*^{-/-} male mice. **g,h**, Magnetic resonance imaging (MRI) analysis of lean mass (**g**) and fat mass (**h**) of *Gpr146*^{+/+} and *Gpr146*^{-/-} littermates on chow diet. * $p < 0.05$, ** $p < 0.01$; bars in **d** indicate mean \pm s.d., bars in **e-h** indicate mean \pm s.e.m..

2. Regarding the energy expenditure, several questions remain. Is the increase in catabolism that was observed in vivo also present in isolated adipocytes? What is the in vivo energy expenditure profile in the adipose specific KOs? Does the increased energy expenditure persist at thermoneutrality? How about on chow? The authors say the mechanism is not through canonical thermogenic cycles but *Adrb3*, which they show is up, is the major driver of both *Ucp1* and non-*Ucp1* dependent thermogenesis. And CL-316,243 does far more than just stimulate lipolysis.

Answer: We thank the reviewer for these thoughtful and constructive questions. To address the first point, we assessed cellular respiration using Seahorse analysis in differentiated human SGBS adipocytes following siRNA-mediated GPR146 knockdown. No significant difference in oxygen consumption rate (OCR) was observed between knockdown and control cells (new data, Fig. R1Q2a), suggesting that the elevated energy expenditure observed in vivo likely involves systemic or multi-tissue interactions not recapitulated in isolated adipocytes.

Furthermore, under HFD feeding condition, we observed a significant increase in energy expenditure in both male and female mice (Fig. 3d, new data, Fig. 3g). However, under chow-fed conditions, we did not observe a significant difference in whole-body energy expenditure between knockout and control mice (new data, Fig. R1Q2b). Interestingly, under chow, knockout mice displayed a significant increase in lean mass (new data, Supplementary Fig. 3g) and respiratory exchange ratio (RER) (new data, Supplementary Fig. 3e). In adipose-specific *Gpr146* knockout mice subjected to HFD feeding, indirect calorimetry similarly revealed no significant change in energy expenditure (new data, Supplementary Fig. 7f). However, we observed a significant increase in RER during the light (inactive) phase (new data, Supplementary Fig. 7g), suggesting a shift toward greater carbohydrate utilization during resting conditions. This altered substrate preference may be secondary to reduced lipid flux from adipose tissue, consistent with lower circulating free fatty acid levels, and aligns with the more modest reduction in body weight and adiposity observed in adipose-specific knockouts compared to global knockouts.

Regarding the mechanistic interpretation, we fully agree with the reviewer that upregulation of *Adrb3* mRNA alone is insufficient to support a definitive role for ADRB3 signalling. In male knockout mice, we did not observe corresponding increases in *Ucp1* expression or other

canonical thermogenic markers, suggesting that UCP1-mediated thermogenesis is unlikely to underlie the increased energy expenditure in males (Supplementary Fig. 6f, g). Interestingly, in female knockout mice, we observed a robust (>15-fold) increase in *Ucp1* expression in eWAT, along with upregulation of *Pgc1α* and *Ppara* (new data, Supplementary Fig. 6h, i), indicating potential activation of UCP1-dependent thermogenesis in females. While *Adrb3* is a known regulator of both UCP1-dependent and -independent thermogenesis, and CL-316,243 can elicit diverse metabolic responses, our current data are limited to mRNA-level changes, which may reflect secondary or compensatory effects. We have revised the manuscript accordingly to avoid overinterpretation.

Finally, we would like to state explicitly that the precise mechanism driving the increased energy expenditure in male *Gpr146* knockout mice remains unknown. We agree with the reviewer that further investigation is required to elucidate the underlying pathways.

Fig.R1Q2, a, Seahorse Mitostress test measuring oxygen consumption rate (OCR) in SGBS mature adipocytes with or without *GPR146* knockdown. **b**, Hourly energy expenditure (EE) in male *Gpr146*^{+/+} and *Gpr146*^{-/-} mice fed chow (n=8 mice per group), averaged over four consecutive days.

Figure 3. GPR146-deficient Mice exhibit Elevated Energy expenditure. g, Indirect calorimetry study in female mice fed HFD for two weeks. Hourly energy expenditure (EE) in female *Gpr146*^{+/+} and *Gpr146*^{-/-} mice fed a high-fat diet, averaged over four consecutive days. Horizontal bars indicate the light (white) and dark (black) photoperiod.

Supplementary Figure 3. GPR146 deficiency reprograms hepatic glucose and amino acid metabolism. e. Respiratory exchange ratio (RER) measured during the dark phase by indirect calorimetry in chow-fed *Gpr146*^{+/+} and *Gpr146*^{-/-} male mice. **g**, Magnetic resonance

imaging (MRI) analysis of lean mass of *Gpr146*^{+/+} and *Gpr146*^{-/-} littermates on chow diet. * *p* < 0.05; bars indicate mean ± s.e.m..

Supplementary Figure 7, Adipose GPR146 mediates protection against diet-induced obesity and liver steatosis, Related to Fig.4. f, Energy Expenditure measured by indirect calorimetry in Adipoq-Cre⁺ and control mice (Adipoq-Cre⁻) fed HFD for 4 months. **g**, Hourly respiratory exchange ratio (RER) measured in the same cohort of male Adipose specific knockout and the control littermates. * *p* < 0.05.

Supplementary Figure 6, Metabolic phenotyping and thermogenic pathway regulation in adipose tissue of GPR146-deficient and control mice, related to Fig3. h, i, qPCR expression analysis of *Ucp1*, *Pgc1a* and *Ppara* in eWAT (**h**) and iWAT (**i**) from female *Gpr146*^{+/+} and *Gpr146*^{-/-} littermates fed HFD (n=4-8 mice per group, by Student's t test). * *p* < 0.05, ** *p* < 0.01; bars indicate mean ± s.d.

3. Why do the author's think that knocking *Gpr146* down in cells has an effect when they haven't added any ligand? Is it known if the ligand for GPR146 is endogenously produced?

Answer: This is an excellent question. There are two plausible explanations for the observed phenotypic effects upon *GPR146* knockdown in the absence of exogenously added ligand.

First, like many GPCRs, GPR146 likely exhibits constitutive (ligand-independent) basal activity, which can influence downstream signalling pathways relevant to adipocyte differentiation. Indeed, we found that knockdown of GPR146 in mature adipocytes reduced ERK activity, whereas over-expression of GPR146 in SGBS cells significantly increased ERK activity even in the absence of serum supplementation (new data, Fig. 8d, supplementary Fig. 11e). This basal activity has been reported for many GPCRs and may regulate intracellular signaling even in the absence of ligand stimulation.

Second, recent studies have identified Cholesin-a gut-derived hormone induced by NPC1L1-mediated cholesterol uptake-as an endogenous ligand for GPR146⁵. While we did not supplement our in vitro differentiation system with Cholesin or other candidate ligands, it is possible that components in the culture medium (e.g., serum lipids or sterol derivatives) or autocrine/paracrine factors secreted by the differentiating cells themselves may serve as endogenous activators of GPR146.

Together, these two possibilities-constitutive receptor activity and potential endogenous ligand availability-offer a mechanistic explanation for why *GPR146* knockdown leads to impaired adipogenesis, even without the addition of an exogenous ligand.

Fig.8

Supplementary Fig.11

Figure 8. GPR146 regulates mature adipocyte lipolysis through ERK signaling. d, Representative Western blots of phosphorylated ERK (p-ERK) and total ERK in SGBS cells with or without doxycycline-induced GPR146 expression.

Supplementary Figure 11, GPR146 regulates mature adipocyte lipolysis through ERK signaling, Related to Fig. 8. e, Representative western blots of p-ERK and ERK in *GPR146* knockdown SGBS-derived mature adipocytes.

4. All the in vivo findings involve prevention of steatosis because of constitutively expressed genetic models. Inducibly depleting *Gpr146* after NAFLD has developed would significantly strengthen the manuscript.

Answer: We thank the reviewer for this important suggestion. We agree that assessing the therapeutic potential of GPR146 depletion after the onset of NAFLD would be a valuable future direction. While we did not perform inducible genetic deletion after NAFLD establishment, we adopted a complementary strategy using an AAV-delivered shRNA system to acutely deplete GPR146 expression in adult C57BL/6 mice together with high-cholesterol NASH diet challenge (new data, Fig. 5a).

Specifically, mice were administered AAVs expressing shRNA targeting *Gpr146*, which resulted in approximately 40% knockdown of *Gpr146* expression in both subcutaneous and visceral white adipose tissues, as well as 80% in the liver (new data, Fig. 5b). Following AAV delivery, mice were fed a MASH-inducing diet for 13 weeks. Notably, *Gpr146* knockdown mice exhibited significantly lower body weight (new data, Fig. 5e), reduced adipose depot size (new data, Fig. 5f, g), decreased hepatic triglyceride accumulation (new data, Fig. 5c), and reduced inflammatory gene expression compared to scramble shRNA controls (new data, Fig. 5d). These findings indicate that acute depletion of GPR146 in adult mice is sufficient to confer protection against diet-induced hepatic steatosis.

While this model was used as a preventative approach, it nonetheless reinforces the translational potential of targeting GPR146 in metabolic disease and provides additional evidence that modulation of its activity in adulthood can yield beneficial effects on adiposity and hepatic lipid accumulation.

Fig. 5

Figure 5. AAV-mediated acute depletion of *Gpr146* confirms its role in promoting hepatic steatosis and adiposity. **a**, Schematic timeline of the AAV cohort. Adult male mice were injected with AAV8 expressing shRNA targeting *Gpr146* (n = 9) or scramble control (n = 7), followed by 13 weeks of feeding with a MASH-inducing diet enriched in cholesterol and fructose. **b**, qPCR expression analysis of *Gpr146* in liver, eWAT, iWAT and BAT to assess knockdown efficiency. **c**, Hepatic triglyceride content in mice with or without *Gpr146* knockdown. **d**, qPCR expression analysis of hepatic expression of key inflammatory genes in mice with or without *Gpr146* knockdown. **e-g**, Body weight (e), iWAT (f), eWAT (g) weights after 13 weeks of MASH diet feeding. * P < 0.05, ** P < 0.01. bars indicate mean ± s.e.m.

Minor

-In the discussion, just because *Ucp1* expression is unchanged does not mean the EE is *Ucp1* independent.

Answer: We thank the reviewer for this helpful clarification. We agree that the absence of *Ucp1* mRNA upregulation alone is not sufficient to conclude that the increased energy expenditure is UCP1-independent. Indeed, UCP1 activity may be regulated post-translational, or may occur in depots not sampled^{6,7}, and we acknowledge that our current data are limited to gene expression analyses. We have revised the discussion accordingly to avoid overinterpretation and to clarify that further protein-level and functional validation will be

required to determine whether UCP1 contributes to the elevated energy expenditure observed in GPR146-deficient male mice.

Reviewer #2 (Remarks to the Author):

In this manuscript, the authors investigated the regulatory role of the G protein-coupled receptor 146 (GPR146) in adipose tissue and liver lipid metabolism. In an *in vivo* model, GPR146 deficiency exhibited a protective effect against obesity induced by a high-fat diet, reducing liver steatosis and accompanying a decrease in circulating liver enzyme levels. Global depletion of GPR146 in mice affected energy homeostasis by increasing energy expenditure during high-fat diet feeding. The study revealed that specific depletion of GPR146 in adipocytes in mice could counteract diet-induced fatty liver by limiting the influx of free fatty acids. Additionally, the authors demonstrated that GPR146 deletion *in vitro* inhibits adipogenesis by blocking the activation of Gαq-PKC AKT signaling.

The subject matter of this study holds considerable significance in the realm of lipid metabolism. Nevertheless, the data outlined in the current version do not entirely substantiate the authors' conclusions. A notable deficiency lies in the absence of a mechanistic explanation detailing how GPR146 influences lipid turnover and adipogenesis. Here are specific comments aimed at potentially enhancing the manuscript.

1. The first question inquires whether the protein expression level of GPR146 is elevated in the adipose tissue of obese individuals compared to non-obese ones.

Answer: We thank the reviewer for this important question. Unfortunately, we were unable to directly assess GPR146 protein levels in adipose tissue of obese versus non-obese individuals due to technical limitations. Like many GPCRs, GPR146 is expressed at low levels *in vivo*, and despite testing multiple commercially available antibodies, we found that while the antibody (ab151626, Abcam) effectively detects recombinantly overexpressed GPR146, it fails to detect endogenous protein, likely due to insufficient abundance of endogenous protein.

However, at the mRNA level, we consistently observed that *GPR146* expression in adipose tissue is inversely correlated with BMI in human cohorts (data adapted from The Adipose Tissue Knowledge Portal⁸) (new data, Fig. R2Q1), and the expression is lower in obese compared to non-obese individuals (new data, Supplementary Fig. 8f). We propose two possible explanations for this discrepancy between function and expression:

1. Compensatory downregulation: The observed reduction in GPR146 expression in obesity may represent a compensatory response to mitigate further adipose expansion or metabolic stress. This phenomenon has been observed in other metabolic regulators that are downregulated in disease states despite their physiological function promoting those same processes.

2. Cell-type composition shift: Obesity induces extensive adipose tissue remodeling, characterized by increased immune cell infiltration, particularly macrophages, and a substantially reduced proportion of mature adipocyte within the depot⁹. Because the expression level of GPR146 in mature adipocytes is much higher than other cell types within the adipose tissue, a relative reduction in adipocyte fraction (or increased inflammatory cell content) could result in an overall decline in measured GPR146 mRNA from bulk tissue.

Additionally, supporting its functional role, adipose *GPR146* expression is also inversely correlated with plasma LDL-C and triglyceride levels in humans (new data, Fig. R2Q1). Importantly, studies from our group and others have clearly demonstrated that genetic deletion of *Gpr146* in mice leads to significant reductions in circulating LDL-C and TG levels. These findings reinforce the notion that reduced *GPR146* expression is functionally protective, even if its mRNA levels in adipose tissue appears downregulated in obesity due to changes in tissue composition.

Fig.R2Q1

Supplementary Fig. 8

f

Figure R2Q1. Correlation of *GPR146* expression in human adipose tissue with BMI (top), and circulating-LDL level (bottom) based on data from the Adipose Tissue Knowledge Portal. **Supplementary Figure 8, *GPR146* expression in adipose tissue is modulated by nutritional and physiological context in mice and humans, Related to Fig.4. f,** Forest plots summarizing *GPR146* expression across multiple human transcriptomic cohorts (data adapted from the Adipose Tissue Knowledge Portal), comparing obese versus non-obese individuals. Source studies included in the meta-analysis are listed alongside each plot.

2. Does the expression level of *GPR146* in wild-type mice also increase over time in various adipose tissues (eWAT, ingWAT, and BAT) during HFD feeding?

Answer: We thank the reviewer for this thoughtful question. In our analysis of wild-type mice subjected to HFD feeding, we observed that *Gpr146* expression was modestly decreased in eWAT, while levels in iWAT and BAT remained largely unchanged compared to chow-fed mice (new data, Supplementary Fig. 8c). Consistently, publicly available datasets also reveal a similar trend, with *GPR146* expression downregulated in adipose tissue of mice after HFD feeding (new data, Supplementary Fig. 8d, e). This expression pattern in mouse adipose tissue is similar as what is observed in humans, as discussed in response to the previous question. As mentioned, we propose two possible explanations for this discrepancy between reduced expression and the protective phenotype associated with *GPR146* deficiency. One possibility is that *Gpr146* downregulation reflects a feedback mechanism during chronic HFD exposure, aimed at limiting further adipocyte expansion or lipid accumulation in metabolically stressed environments. Alternatively, changes in adipose tissue composition during obesity, including increased infiltration of immune and stromal cells, may reduce the relative abundance of *Gpr146*-expressing mature adipocytes, thereby contributing to the observed decline in total mRNA levels from bulk tissue.

Taken together, these observations suggest that although *GPR146* expression may decline under chronic obesity, its activity is likely important in regulating early adipocyte differentiation and lipid handling. Thus, loss of *GPR146* function- either genetically or through targeted inhibition- can still provide metabolic protection, especially under HFD feeding.

Supplementary Fig.8

Supplementary Figure 8, GPR146 expression in adipose tissue is modulated by nutritional and physiological context in mice and humans, Related to Fig.4. **c**, *Gpr146* mRNA expression in eWAT, iWAT, and BAT from WT C57/BL6 mice fed a chow or high-fat diet (HFD) for 3 months (n = 6–10 per group). **d**, **e**, Analysis of mouse eWAT transcriptomic data from two publicly available datasets, GSE39549 (**d**) and GSE97271 (**e**), comparing *Gpr146* expression before and after HFD feeding, and its relationship with *Lep* (leptin) expression. * $p < 0.05$, ** $p < 0.01$; bars indicate mean \pm s.e.m,

3. Global depletion of GPR146 in mice, as observed in Fig 2b by MRI measurements, led to an increase in the percentage of lean mass under HFD feeding. However, the data only indicate the percentage of lean mass. Is there an increase in the absolute amount of lean mass as well? Utilizing a whole-body composition analyzer might offer more precise *in vivo* measurements the mass of body lean, fat, and fluid.

Answer: We appreciate the reviewer's suggestion regarding the assessment of absolute lean mass. In our study, body composition was measured using MRI, which provided both relative and absolute values for lean and fat mass.

While we observed a significant increase in the percentage of lean mass in *Gpr146* knockout mice under HFD feeding (Fig. 2b), the absolute lean mass did not differ significantly between knockout and control groups (Fig. 3a). The increased percentage is therefore a relative change, driven primarily by the reduction in fat mass rather than a gain in lean tissue under HFD feeding.

4. Does the protective role of GPR146 deficiency against high-fat diet (HFD)-induced obesity suggest a contrasting effect when GPR146 is up-regulated? The authors should provide or at least discuss the results induced by GPR146 over-expressions.

Answer: We thank the reviewer for this thoughtful question. While it may be intuitive to expect that *GPR146* overexpression (OE) would elicit phenotypes opposite to those seen in *GPR146* knockout (KO) models- such as increased adiposity and worsened metabolic outcomes- our findings suggest that the relationship is not strictly linear.

In our *in vitro* studies using human SGBS preadipocytes, we observed that both *GPR146* knockdown (KD) and overexpression (OE) impaired adipocyte differentiation (Fig. 6i, j, k and new data, Supplementary Fig. 9h, i). This suggests that tight regulation of *GPR146* expression is critical for proper adipogenesis, and that perturbation in either direction can be detrimental.

This concept is not without precedent. For example, the MAPK/ERK signalling pathway, a well-studied regulator of adipogenesis, also demonstrates phase-specific and finely tuned control during adipogenesis. ERK activation is required to initiate preadipocyte commitment, but sustained ERK activation inhibits differentiation, and timely inactivation is necessary for

late stage of adipogenesis¹⁰. This illustrates how precise modulation, rather than unidirectional activation or suppression, is essential for adipogenesis.

We have incorporated these new data into the revised manuscript and included a discussion of this observation. Taken together, our findings indicate that GPR146 expression likely needs to be maintained within an optimal range for normal adipocyte function. We now emphasize in the discussion that GPR146 modulation exhibits non-linear effects, and that both gain- and loss-of-function perturbations can disrupt adipogenesis, cautioning against the assumption of symmetric phenotypic outcomes.

Supplementary Fig.9

Supplementary Figure 9, GPR146 promotes differentiation of both mouse and human preadipocytes, Related to Fig.6. h, Relative *GPR146* mRNA expression in human SGBS cells following doxycycline-induced overexpression of *GPR146* ($n = 3$ replicates per condition, representative of 3 independent experiments; Student's t test). i, Triglyceride content in SGBS-differentiated adipocytes overexpressing *GPR146* or GFP (control). * $p < 0.05$, ** $p < 0.01$; bars indicate mean \pm s.e.m.

5. Moreover, the data from GPR146 deficiency do not entirely support the authors' conclusions regarding the promoting effects of GPR146 on adipogenesis. To strengthen this assertion, the author should include data from GPR146 overexpression experiments.

Answer: We thank the reviewer for raising this important point. As demonstrated in our study, both genetic knockout of GPR146 in mouse SVF cells and siRNA-mediated knockdown in human SGBS preadipocytes consistently resulted in impaired adipogenesis, supporting a physiological role for GPR146 in promoting adipocyte differentiation (Fig.6 i-k, Supplementary Fig.9d, e)

As discussed in the previous question, we also performed GPR146 overexpression (OE) experiments in SGBS cells. Interestingly, similar to knockdown, GPR146 OE also inhibited adipogenesis, suggesting that both insufficient and excessive receptor activity can disrupt the differentiation process (new data, Supplementary Fig.9h, i). These findings imply that GPR146 expression must be tightly regulated, and deviations in either direction can be detrimental—consistent with the concept of a dose-sensitive or biphasic effect, which is not uncommon in GPCR signalling. That said, we place greater physiological weight on the loss-of-function (LOF) data *in vivo*, as the knockout and knockdown models show coherent metabolic and cellular phenotypes under both normal and high-fat diet conditions.

We have now included a brief discussion of the OE findings in the revised manuscript and clarified that while OE results do not produce an opposite phenotype, they reinforce the conclusion that appropriate levels of GPR146 are critical for adipose tissue development.

Supplementary Fig.9

Supplementary Figure 9, GPR146 promotes differentiation of both mouse and human preadipocytes, Related to Fig.6. h, Relative *GPR146* mRNA expression in human SGBS cells following doxycycline-induced overexpression of *GPR146* ($n = 3$ replicates per condition, representative of 3 independent experiments; Student's *t* test). i, Triglyceride content in SGBS-differentiated adipocytes overexpressing *GPR146* or GFP (control). * $p < 0.05$, ** $p < 0.01$; bars indicate mean \pm s.d.,

6. The author discussed ATP-consuming futile cycles in whole-body GPR146 deficiency. It raises the question whether similar findings could be observed in mice with adipocyte- or hepatocyte-specific GPR146 deficiency.

Answer: We thank the reviewer for raising this important point. We have examined the expression of genes associated with ATP-consuming futile cycles in adipose tissue of adipocyte-specific *Gpr146* knockout mice. In both eWAT and ingWAT depots, we did not observe significant changes in the mRNA levels of key genes involved in lipid cycling pathways (new data, Fig. R2Q6a, b). Additionally, in this revision, we also performed indirect calorimetry in adipose-specific KO mice and found no significant difference in energy expenditure, despite a clear reduction in fat mass (new data, Supplementary Fig. 7e, f). One possible explanation is that the metabolic phenotype in adipocyte-specific knockout mice is considerably smaller than that observed in whole-body knockout mice (e.g., ~9% vs ~16% reduction in body weight after 8 weeks of HFD feeding), which might limit our ability to detect changes at the transcriptomic level.

As for liver-specific *Gpr146* knockout mice, we did not observe any significant differences in body weight or hepatic triglyceride levels under HFD feeding conditions (supplementary Fig. 7a and Fig. 4C). Given the lack of a clear metabolic phenotype, we did not proceed with further analysis of futile cycle gene expression in adipose tissue from these mice. It is worth noting, however, that our previous study¹¹ demonstrated that GPR146 regulates plasma cholesterol and triglyceride levels primarily through its function in the liver, particularly via modulation of hepatic VLDL secretion. In contrast, the regulation of hepatic lipid accumulation in response to HFD feeding appears to be predominantly governed by GPR146 activity in adipose tissue, rather than in hepatocytes.

We also acknowledge the reviewer's concern regarding our original interpretation of the futile lipid cycle as a mechanistic driver of increased energy expenditure. As we now clarify in the revised manuscript, this conclusion was based primarily on qPCR data showing increased expression of genes involved in lipid turnover and re-esterification. We agree that these findings are not sufficient to establish causality, and such changes could reflect secondary adaptations rather than a direct mechanistic pathway. We have therefore revised the relevant sections to tone down the overinterpretation and relocated the discussion of lipid cycling to the Discussion section, where we clearly state that the precise mechanism underlying increased resting energy expenditure in male *Gpr146* knockout mice remains unresolved.

In our revised study, we also performed indirect calorimetry in female *GPR146*-deficient mice, and found that KO females exhibited significantly increased energy expenditure compared to controls during both light and dark cycles (new data, Fig. 3g). Interestingly, in contrast to male KO mice where *Ucp1* expression remained unchanged in eWAT and ingWAT (Supplementary Fig. 6f, g), female KO mice displayed a more than 15-fold increase in *Ucp1* expression specifically in eWAT, along with marked upregulation of *Pgc1 α* and *Ppara* (new data, Supplementary Fig. 6h, i). These data suggest that in females, activation of UCP1-mediated thermogenesis may contribute to the observed increase in energy expenditure, potentially through enhanced mitochondrial biogenesis and oxidative metabolism. This emerging sex difference in thermogenic activation may reflect differential hormonal regulation or adipose tissue plasticity between males and females, and further investigation to elucidate sex-specific mechanisms underlying GPR146-dependent metabolic control is required.

To further explore the role of acute *GPR146* depletion, we performed an AAV-shRNA-mediated knockdown of *Gpr146* in wild-type mice. This approach led to efficient knockdown in the liver (~80% mRNA reduction) and moderate knockdown (~40%) in white adipose tissues (new data, Fig. 5b). Notably, mice with *Gpr146* depletion via AAV-shRNA exhibited significantly reduced body weight, decreased eWAT and ingWAT depot sizes, and lower hepatic triglyceride content under HFD feeding (new data, Fig. 5e, f, g, c). These results support the notion that partial and acute reduction of GPR146 in both liver and adipose tissue is sufficient to confer metabolic benefits, and align well with the phenotypes observed in our genetic knockout models.

Fig. R2Q6**Supplementary Fig.7****Fig. R2Q6****Fig.3****Supplementary Fig.6****Supplementary Fig.6****Fig.5****Fig.5**
Fig.R2Q6. a,b. Relative mRNA expression of genes involved in lipid cycling in iWAT (a) and eWAT (b) from adipo-specific knockout mice and their control littermates fed HFD for 4 months. * $p < 0.05$, ** $p < 0.01$; bars indicate mean \pm s.d.,

Supplementary Figure 7, Adipose GPR146 mediates protection against diet-induced obesity and liver steatosis, Related to Fig.4. e, MRI analysis of fat mass and lean mass from female adipose-specific knockout mice ($Adipoq-Cre^+$) and control littermates ($Adipoq-Cre^-$) fed HFD for 4 months. **f,** Energy Expenditure measured by indirect calorimetry in $Adipoq-Cre^+$ and control mice ($Adipoq-Cre^-$) fed HFD for 4 months. * $p < 0.05$, ** $p < 0.01$; bars in all panels indicate mean \pm s.d..

Figure 3. GPR146-deficient Mice exhibit Elevated Energy expenditure. g, Indirect calorimetry study in female mice fed HFD for two weeks. Hourly energy expenditure (EE) in female $Gpr146^{+/+}$ and $Gpr146^{-/-}$ mice fed a high-fat diet, averaged over four consecutive days. Horizontal bars indicate the light (white) and dark (black) photoperiod (n=8 mice per group).

Supplementary Figure 6, Metabolic phenotyping and thermogenic pathway regulation in adipose tissue of GPR146-deficient and control mice, Related to Fig.3. h,i, qPCR expression analysis of *Ucp1*, *Pgc1a* and *Ppara* in eWAT (h) and iWAT (i) from female

Gpr146^{+/+} and *Gpr146*^{-/-} littermates fed HFD (n=4-8 mice per group, by Student's t test). * $p < 0.05$, ** $p < 0.01$; bars in h and i indicate mean \pm s.d.; bars in f and g indicate mean \pm s.e.m..

Figure 5. AAV-mediated acute depletion of Gpr146 confirms its role in promoting hepatic steatosis and adiposity. **b**, qPCR expression analysis of *Gpr146* in liver, eWAT, iWAT and BAT to assess knockdown efficiency. **c**, Hepatic triglyceride content in mice with or without *Gpr146* knockdown. **e-g**, Body weight (**e**), iWAT(**f**), eWAT (**g**) weights after 13 weeks of MASH diet feeding. * $P < 0.05$, ** $P < 0.01$. bars indicate mean \pm s.e.m.

7. Since PKC is the endpoint of the proposed GPR146 signaling pathway, in the GPR146 depletion background, inhibition of PKC function by pharmacological reagent alone cannot support the authors' hypothesis that Gαq-PKC-pAkt is downstream of GPR146 signaling.

Answer: We thank the reviewer for their thoughtful question. However, we would like to clarify that in our proposed signalling pathway, AKT (measured as p-AKT), not PKC, is the downstream endpoint. Our hypothesis states that GPR146 activates Gαq, which in turn activates PKC, leading to the phosphorylation of AKT (p-AKT), a key effector in subcutaneous adipogenesis. Thus, we assess pathway activity by measuring p-AKT levels, not PKC directly.

In our experiments, GPR146 knockdown (KD) pre-adipocytes exhibited reduced p-AKT levels, indicating that the pathway is compromised when GPR146 is depleted. When we treated these KD cells with a PKC inhibitor, we observed no further reduction in p-AKT or adipogenesis, unlike in control cells where the inhibitor significantly decreased both. This lack of additive effect in KD cells supports our model: GPR146 depletion upstream already disrupts the Gαq-PKC-pAkt cascade, so inhibiting PKC downstream has no additional impact. In contrast, PKC inhibition in control cells confirms its role in the intact pathway (Fig. 7f, g, h).

Therefore, our use of p-AKT as a readout, combined with the PKC inhibitor experiments, directly supports the hypothesis that Gαq-PKC-pAkt operates downstream of GPR146. We hope this clarifies that AKT, not PKC, is the endpoint we are evaluating.

8. To address the roles of GPR146 in lipid turnover, the authors should provide more functional experiments, instead of just differential expressions analyses.

Answer: We appreciate the reviewer's suggestion to include additional functional evidence beyond transcriptomic analyses to support our claim that GPR146 regulates lipid turnover. We fully agree that our initial interpretation may have overstated the mechanistic link between GPR146 deficiency and increased lipid turnover, particularly given that the original conclusion was based primarily on differential gene expression. Honestly, while we present robust data demonstrating that adipose-specific depletion of GPR146 attenuates both adipogenesis and lipolysis-thereby reducing FFA efflux and contributing to protection against hepatic steatosis-the precise molecular mechanism by which GPR146 deficiency enhances energy expenditure remains unclear in male mice. Although we initially speculated the involvement of ATP-consuming futile cycles, our evidence is limited to mRNA expression data. In the revised manuscript, we have moved this hypothesis into the Discussion section and clearly acknowledged the current limitations in direct mechanistic evidence.

In our revised study, we also performed indirect calorimetry in female GPR146-deficient mice, and found that KO females exhibited significantly increased energy expenditure compared to controls during both light and dark cycles (new data, Fig. 3g). Interestingly, in contrast to male KO mice where *Ucp1* expression remained unchanged in eWAT and ingWAT (Supplementary Fig. 6f, g), female KO mice displayed a more than 15-fold increase in *Ucp1* expression

specifically in eWAT, along with marked upregulation of *Pgc1α* and *Ppara* (new data, Supplementary Fig. 6h, i). These data suggest that in females, activation of UCP1-mediated thermogenesis likely contribute to the observed increase in energy expenditure, potentially through enhanced mitochondrial biogenesis and oxidative metabolism. This emerging sex difference in thermogenic activation may reflect differential hormonal regulation or adipose tissue plasticity between males and females^{1,2}, and further investigation to elucidate sex-specific mechanisms underlying GPR146-dependent metabolic control is required.

Importantly, we would like to highlight the systemic metabolic reprogramming in GPR146-deficient mice. The knockout mice exhibit improved glucose handling, accompanied by increased hepatic glycogen content and elevated levels of hexose and pentose phosphate intermediates, suggesting a possible enhancement in hepatic insulin sensitivity (Fig. 2k, l and Supplementary data Fig. 3a, b, c). The accumulation of pentose phosphate intermediates further suggests increased NADPH production³, which could contribute to redox homeostasis and protection from oxidative stress, a key driver of hepatic inflammation in NAFLD⁴.

Additionally, untargeted metabolomics also revealed significantly elevated hepatic levels of seven standard amino acids in *Gpr146*^{-/-} mice, namely histidine, cysteine, lysine, valine, leucine, isoleucine, and phenylalanine (new data, Supplementary Fig. 3d). Consistent with these hepatic metabolic changes, chow-fed *Gpr146*^{-/-} mice exhibited a significantly higher respiratory exchange ratio (RER) during the dark cycle compared to controls (new data, Supplementary Fig. 3e, f), indicating a greater reliance on glucose as an energy source. Furthermore, chow-fed *Gpr146*^{-/-} mice displayed significantly increased lean mass without changes in fat mass (new data, Supplementary Fig. 3g, h), potentially reflecting altered systemic nutrient partitioning and increased availability of amino acids at chow-fed baseline condition. Together, these findings support a model in which GPR146 deficiency induces coordinated metabolic shifts that reduce the need for lipid storage and limit ectopic lipid accumulation in the liver.

To directly probe the role of GPR146 in mature adipocyte where it is most highly expressed, we performed complementary *in vitro* and *in vivo* lipolysis assays. In mature SGBS-derived adipocytes, we found that doxycycline-induced GPR146 overexpression significantly increased lipolysis, as measured by glycerol release (new data, Fig. 8c). Furthermore, we demonstrated that this increase in lipolysis was dependent on ERK signalling as treatment with the MEK1 inhibitor PD0325901 attenuated the lipolysis response (new data, Fig. 8c). In contrast, siRNA-mediated knockdown of *GPR146* in SGBS adipocytes significantly reduces lipolytic activity (new data, Fig. 8b).

To assess physiological relevance, we utilized an AAV-shRNA-based knockdown approach to transiently deplete GPR146 *in vivo* in mice fed a NASH diet. GPR146 knockdown led to significant reductions in body weight, adipose depot mass, and hepatic triglyceride levels, closely mirroring the phenotypes observed in our constitutive knockout models (new data, Fig. 5c, e, f, g). Importantly, upon CL-316,243 stimulation, glycerol release was reduced in GPR146 knockdown mice compared to controls, suggesting diminished lipolytic capacity, which likely contributes to the observed reduction in hepatic TG accumulation (new data, Fig. 8e).

While these new findings do not establish a direct role for GPR146 in driving lipid turnover through a futile cycle, they support a functional link between GPR146 activity in adipose tissue and hepatic lipid homeostasis, particularly by modulating lipolysis and FFA availability to the liver. We have incorporated these new data into the revised manuscript to address the reviewer's comment and strengthen our conclusions.

Figure 3. GPR146-deficient Mice Have Elevated Energy expenditure during Light Photoperiod. **g**, Indirect calorimetry study in female mice fed HFD for two weeks. Hourly energy expenditure (EE) in female *Gpr146*^{+/+} and *Gpr146*^{-/-} mice fed a high-fat diet, averaged over four consecutive days. Horizontal bars indicate the light (white) and dark (black) photoperiod (n=8 mice per group). * $p < 0.05$, ** $p < 0.01$.

Supplementary Figure 6, Metabolic phenotyping and thermogenic pathway regulation in adipose tissue of GPR146-deficient and control mice, related to Fig3. **h,i**, qPCR expression analysis of *Ucp1*, *Pgc1a* and *Ppara* in eWAT (**h**) and iWAT (**i**) from female *Gpr146*^{+/+} and *Gpr146*^{-/-} littermates fed HFD (n=4-8 mice per group, by Student's t test). * $p < 0.05$, ** $p < 0.01$; bars in **h** and **i** indicate mean \pm s.d.; bars in **f** and **g** indicate mean \pm s.e.m.. **Supplementary Figure 3. GPR146 deficiency reprograms hepatic glucose and amino acid metabolism.** **d**, Relative abundance of hepatic amino acids in 16 h fasted male mice fed an HFD (n=7 mice per group, by Student's t test). **e,f**, Respiratory exchange ratio (RER) measured during the dark (**e**) and light (**f**) phases by indirect calorimetry in chow-fed *Gpr146*^{+/+} and *Gpr146*^{-/-} male mice. **g,h**, Magnetic resonance imaging (MRI) analysis of lean mass (**g**) and fat mass (**h**) of *Gpr146*^{+/+} and *Gpr146*^{-/-} littermates on chow diet. * $p < 0.05$, ** $p < 0.01$; bars in **a-d** indicate mean \pm s.d., bars in **e-h** indicate mean \pm s.e.m..

Figure 8. GPR146 regulates mature adipocyte lipolysis through ERK signaling. **b**, Glycerol release from SGBS mature adipocytes transfected with siGPR146 or control siRNA, with or without forskolin stimulation; values normalized to intracellular triglyceride (n=3 replicates per experiment, representative of 3 independent experiments, by Student's t test). **c**, Glycerol release from SGBS adipocytes with doxycycline-inducible GPR146 expression. Measurements were performed with or without the MEK inhibitor PD98059 and normalized to intracellular TG (n=3 replicates per experiment, representative of 3 independent

experiments, by Student's t test). **e**, In vivo lipolysis assay following CL-316,243 stimulation in mice injected with AAV-sh*Gpr146* or scramble control shRNA and maintained on a MASH-inducing diet for 6 weeks. Plasma glycerol was measured at the indicated time points upon CL316,243 injection. * $p < 0.05$, ** $p < 0.01$; bars indicate mean \pm s.e.m..

Figure 5. AAV-mediated acute depletion of Gpr146 confirms its role in promoting hepatic steatosis and adiposity. **c**, Hepatic triglyceride content in mice with or without *Gpr146* knockdown. **e-g**, Body weight(**e**), iWAT(**f**), eWAT (**g**) weights after 13 weeks of MASH diet feeding. * $P < 0.05$, ** $P < 0.01$. bars indicate mean \pm s.e.m.

Minor points:

1. It would be helpful to provide a brief description of G protein-coupled receptors in mediating functional activities of adipose tissues.

Answer: Among the molecular pathways orchestrating adipose tissue function, G protein-coupled receptors (GPCRs) play a central role in regulating lipid metabolism, adipogenesis, and systemic energy balance¹². Several GPCRs expressed in adipose tissue, such as the cannabinoid receptor CB1, have been shown to promote hepatic steatosis by increasing adipocyte lipogenesis and FFAs released into circulation. Conversely, adipose-specific deletion of CB1 protects against hepatic lipid accumulation. These findings highlight the essential role of GPCR-mediated adipose signaling in coordinating systemic lipid flux and position adipose-resident GPCRs as important contributors to MASLD pathophysiology.

We've included this paragraph in the revised manuscript.

2. The image should include a scale bar for reference.

Answer: We've included scale bar as suggested.

3. Please include the sequences for the siRNA used.

Answer: siRNA sequences were provided as suggested.

Reviewer #3 (Remarks to the Author):

The study by Shi et al. provides a large set of data investigating the effects of GPR146 on liver steatosis and particularly the role of adipocyte GPR146. The authors first describe findings extracted from public GWAS databases indicating association of GPR146 variants with increased plasma liver enzymes and CRP. In the following experiments they show that GPR146 knockout reduces liver steatosis and plasma ALT in mice. Liver metabolomics was used to characterize in more detail effects on lipids. GPR146 knockout also decreased weight gain and fat mass in response to HFD that appears to be related to elevated energy expenditure during resting periods. The authors used tissue specific knockout models to demonstrate that Adipoq-Cre but not Alb-Cre induced knockout alleviated the liver steatosis suggesting that adipocyte GPR146 mediates the steatotic effect. Adipocyte KO of GPR146 also reduced weight gain, fat mass and plasma FFA. In cell models GRP146 was found to promote adipogenesis potentially through Gαq-PCK-AKT pathway. The study is interesting and thorough. However, there are several points of concern:

1. The effect of Adipoq-Cre induced knockout on Grp146 expression in WAT requires further

information. At least after 4 months of HFD the effect of the KO on *Gpr146* expression appears either limited (ingWAT) or not existing (eWAT) (Extended Figure 7e). Please provide necessary evidence of the effect of *Gpr146* KO on GPR146 expression in different time points used in the study. The current data raises questions about the mechanisms of the observed effects in Adipoq-KO mice.

Answer: We thank the reviewer for this important observation. We agree that the limited *Gpr146* depletion in eWAT after 4 months of HFD feeding in the Adipoq-Cre-driven adipose-specific knockout model raises valid questions about the interpretation of the observed phenotypes.

We acquired the Adipoq-Cre line from The Jackson Laboratory, and proper genotyping was confirmed prior to initiating experiments. In young adult mice (8 weeks old) maintained on chow diet, we validated that *Gpr146* expression was significantly reduced across multiple adipose depots, including iWAT, eWAT, and BAT, but not in the liver and muscle, in Adipoq-Cre+;*Gpr146*^{fl/fl} mice compared to Cre-negative littermate controls (new data, Supplementary Fig. 7d). This confirms that Cre-mediated recombination is effective.

However, as the reviewer correctly noted, after prolonged HFD feeding (approximately 4 months), *Gpr146* expression in eWAT was no longer significantly different between knockout and control mice (Supplementary Fig. 7c), whereas iWAT still showed consistent and significant depletion. The reason for this apparent loss of knockdown efficiency over time in eWAT is not fully understood. One possible explanation is the regional differences in Cre recombination efficiency under prolonged metabolic stress. Additionally, prolonged HFD feeding is known to cause substantial changes in adipose tissue composition where there is marked infiltration of immune cells and a corresponding reduction in the proportion of mature adipocytes⁹—the cell type in which Adipoq-Cre is active. This deficit in mature adipocytes could dilute the overall depletion effect observed at the mRNA level when analysing bulk tissue.

We acknowledge that this temporal limitation complicates the interpretation of long-term adipose-specific KO phenotypes. As such, we have revised the manuscript to discuss this caveat and have more cautiously interpreted the Adipoq-Cre results.

Supplementary Fig.7

Supplementary Figure 7, Adipose GPR146 mediates protection against diet-induced obesity and liver steatosis, Related to Fig.4. d, Relative expression of *Gpr146* mRNA in iWAT, eWAT, BAT, liver and muscle from male mice at 8-week-old (n=8 mice per group, by Student's t test). * $p < 0.05$, ** $p < 0.01$; bars indicate mean \pm s.d..

2. The authors state in several instances that GPR146 KO ameliorates hepatic inflammation. However, I don't think any evidence is presented to support this notion. ALT cannot be considered and an inflammation marker. However, indeed it would be necessary to study the role of GPR146 in relation to liver inflammation and fibrosis. Simple HFD model may not be the best for this purpose, but it may require additional factors such as fructose feeding. The

authors also reported elevated CRP associated with GPR146 variants in humans. What about in mice? Does the Gpr146 KO affect CRP?

Answer: We thank the reviewer for highlighting this important point. We agree that ALT is a marker of hepatocellular injury, but not a specific indicator of hepatic inflammation. In our study, we performed transcriptomic analysis of liver tissue from both male and female *Gpr146* knockout and control mice under HFD conditions. Gene set enrichment analysis (GSEA) revealed significant downregulation of multiple inflammation-related pathways, including *inflammatory response*, *TNF α signaling via NF- κ B*, and *interferon- γ response*, in *Gpr146*-deficient mice compared to controls. These findings suggest a reduction in hepatic inflammatory tone in the absence of GPR146 (Fig. 1l ,m).

To further support this observation, we evaluated inflammatory gene expression in an independent model using AAV-shRNA-mediated knockdown of *Gpr146* in adult mice fed a high-cholesterol, fructose-containing NASH diet, which better models liver inflammation¹³. In this model, we observed significant reductions in hepatic mRNA levels of key pro-inflammatory cytokines, including *Tnfa*, *Il1 β* , and *Il6*, in *Gpr146* knockdown mice relative to controls (new data, Fig. 5d). While these findings support a consistent pattern of reduced inflammatory gene expression, we recognize the reviewer's concern regarding overinterpretation. Accordingly, we have carefully revised the manuscript to more precisely state that GPR146 depletion protects against hepatic triglyceride accumulation, and we now explicitly clarify that our data demonstrate a downregulation of inflammatory gene expression, rather than direct evidence of reduced hepatic inflammation.

Regarding CRP, while our human eQTL analysis suggests that GPR146 variants are associated with altered CRP levels¹⁴, we did not observe a consistent change in hepatic *Crp* mRNA level or plasma CRP concentration in *Gpr146* knockout mice under HFD conditions. This may reflect species-specific differences in CRP regulation, as CRP is a much stronger acute-phase reactant in humans than in mice (new data, Fig. R3Q2 a, b).

Fig.5

Fig.R3Q2

Figure 5. AAV-mediated acute depletion of Gpr146 confirms its role in promoting hepatic steatosis and adiposity. d, qPCR expression analysis of hepatic expression of key inflammatory genes in mice with or without *Gpr146* knockdown. * $P < 0.05$, ** $P < 0.01$. bars indicate mean \pm s.e.m.

Fig.R3Q2. a, Relative expression of *Crp* in liver of *Gpr146*^{+/+} and *Gpr146*^{-/-} mice littermates fed HFD for 3 months (n=4 per group, by Student's t test). **b,** Plasma concentration of CRP estimated by CPR ELISA. * $P < 0.05$, ** $P < 0.01$. bars indicate mean \pm s.e.m.

3.The effect on glucose metabolism should be better explained. Do the increased liver hexose metabolites indicate upregulation of hepatic glucose uptake or what is the authors'

interpretation? It is not clear what the authors mean by statement that “GRP146 deficiency promotes glucose mobilization”, page 5 line 9-10. Indeed, glycogen storage was increased.

Answer: We thank the reviewer for this insightful comment and the opportunity to clarify our interpretation regarding glucose metabolism in GPR146-deficient mice. The original statement that “GPR146 deficiency promotes glucose mobilization” was imprecise, and we agree it could be misinterpreted as an increase in glucose output or utilization. We have revised this in the manuscript to improve clarity and accuracy.

In our study, *Gpr146* knockout mice exhibited significantly improved glucose and insulin tolerance tests, consistent with enhanced systemic insulin sensitivity. In the liver, we observed a notable increase in glycogen content as well as elevated levels of hexose and pentose phosphate intermediates. While we cannot conclusively determine whether the glycogen accumulation results from increased glucose uptake or reduced glucose oxidation, these findings suggest that carbohydrate flux is redirected toward storage in the liver, likely attributed to enhanced hepatic insulin sensitivity in the absence of GPR146.

To further explore this, we performed indirect calorimetry in chow-fed mice and found that GPR146-deficient mice exhibited significantly enhanced respiratory exchange ratio (RER) (new data, Supplementary Fig. 3e), suggesting a greater reliance on carbohydrates as an energy source. When considered alongside the increased hepatic glycogen content (Supplementary Fig. 3b, c) and significantly increased lean mass (new data, Supplementary Fig. 3g), these findings suggest that, in GPR146-KO mice under chow, nutrient partitioning is reprogrammed: glucose is preferentially used for energy and lean tissue maintenance rather than contributing to lipogenesis in liver and adipose tissue. We propose that this metabolic shift results from a combination of enhanced hepatic insulin sensitivity and a defect in adipogenesis, leading to reduced lipid storage capacity and altered systemic energy allocation.

This metabolic profile is consistent with improved hepatic insulin action, which drives glucose storage as glycogen and promotes diversion of glucose into the pentose phosphate pathway. Importantly, the accumulation of pentose phosphate pathway intermediates suggests increased NADPH production (Supplementary Fig. 3a), which may serve a cytoprotective role by maintaining redox homeostasis and mitigating oxidative stress. Given the well-established link between oxidative stress and hepatic inflammation in NAFLD progression, we think that enhanced PPP flux may contribute to the anti-inflammatory and hepatoprotective effects observed in GPR146-deficient mice.

This metabolic shift—characterized by reduced lipid accumulation, enhanced glucose storage, and increased redox buffering capacity—combined with diminished FFA influx into the liver, offers a coherent explanation for the observed reduction in hepatic triglycerides, improved insulin sensitivity, and attenuated inflammatory gene expression. We have revised the manuscript accordingly to reflect this mechanistic insight and thank the reviewer for prompting this clarification.

Supplementary Fig.3

Supplementary Figure 3. GPR146 deficiency reprograms hepatic glucose and amino acid metabolism. e., Respiratory exchange ratio (RER) measured during the dark phase by

indirect calorimetry in chow-fed *Gpr146*^{+/+} and *Gpr146*^{-/-} male mice. **g**, Magnetic resonance imaging (MRI) analysis of lean mass of *Gpr146*^{+/+} and *Gpr146*^{-/-} littermates on chow diet. * $p < 0.05$, ** $p < 0.01$; bars indicates mean \pm s.e.m..

4. The authors claim that GRP146 KO ameliorates insulin resistance. However, this is not obvious. The KO improves glucose tolerance (fig. 2g) which could suggest better insulin tolerance. However, in ITT the effects can be seen only in late time points when the animals are recovering from the insulin effect. A true difference in insulin sensitivity should be seen in early time points when glucose is decreasing. This is especially obvious in female mice where the effect in ITT is observed only after 90 min (Ex. Fig. 3d) but also to some extent in male mice (Fig 2h). The results rather appear to suggest defects in counter-regulatory mechanisms correcting insulin-induced hypoglycemia. Also, there was no difference in plasma fasting insulin (Ex. Fig. 3e) again suggesting no clear effect on insulin sensitivity. HOMA-IR index, preferably calculated with the mouse modified formula (doi: 10.1177/0023677212473714) could be provided also.

Answer: We appreciate the reviewer's critical evaluation of our claim regarding insulin sensitivity. The improved glucose tolerance observed in *Gpr146* knockout (KO) mice (Fig. 2k) initially suggested enhanced insulin responsiveness. However, we agree that the insulin tolerance test (ITT) results do not clearly support this interpretation. As the reviewer rightly pointed out, the differences in blood glucose during ITT are primarily observed during the late recovery phase (after 60–90 minutes), particularly in female mice (Supplementary Fig. 5b) and to some extent in males (Fig. 2l), rather than during the early glucose-lowering phase where true insulin sensitivity is best assessed. Consistently, glucose tolerance tests (GTT) showed a similar temporal pattern: *Gpr146* knockout mice exhibited lower blood glucose levels mainly during the later stages of the test (Fig. 2k and Supplementary Fig. 5a), without significant differences in the early glucose clearance phase. Taken together, these findings point toward improved hepatic glucose handling and metabolic flexibility in *Gpr146*-deficient mice, rather than a direct enhancement of peripheral insulin sensitivity¹⁵.

Furthermore, we observed no significant differences in fasting insulin levels between *Gpr146* KO and control mice (Supplementary Fig. 5d), and this is consistent with the possibility that systemic insulin sensitivity is not markedly improved. Following the reviewer's suggestion, we have now calculated the HOMA-IR index using fasting glucose and insulin levels, applying the mouse-specific formula¹⁶. These results are now included in the revised manuscript (**new data, Supplementary Fig. 5e**) to provide an additional and more robust assessment of insulin resistance.

In light of these data, we have revised our manuscript to avoid overclaiming improved insulin sensitivity. Instead, we now describe the metabolic phenotype of *Gpr146* KO mice as reflecting a complex reprogramming of hepatic and systemic metabolism. Notably, liver metabolomics revealed increased glycogen storage, reduced lipid species (including FFAs and long-chain acyl-CoAs), and elevated pentose phosphate intermediates, suggesting a shift toward glucose storage and redox balancing (Fig. 1h, i, and supplementary Fig. 3a-c).

In support of this view, transient *Gpr146* knockdown mice showed reduced glycerol release in response to CL-316,243 stimulation, indicating diminished lipolytic activity (**new data, Fig. 8e**). We hypothesize that this reduction in circulating glycerol may impair hepatic gluconeogenesis, contributing to the blunted counter-regulatory response during the ITT recovery phase.

Taken together, these findings suggest that GPR146 deficiency induces a metabolic state characterized by altered glucose handling, reduced substrate availability for gluconeogenesis, and a preference for non-lipid oxidative pathways, rather than classical improvements in

insulin sensitivity. We have revised the manuscript accordingly to reflect this interpretation and thank the reviewer for prompting this clarification.

Supplementary Fig.5

Fig.8

Supplementary Figure 5. GPR146 deficiency alters systemic glucose metabolism, Related to Fig.2. e, HOMA-IR calculated from fasting glucose and insulin levels in mice fed HFD for 1 month (n = 8–11 per group; Student’s *t*-test). * *p* < 0.05, ** *p* < 0.01; bars in a, b and e indicate mean ± s.e.m.; bars in d indicate mean ± s.d..

Figure 8. GPR146 regulates mature adipocyte lipolysis through ERK signaling. e, *In vivo* lipolysis assay in mice with acute depletion of *Gpr146* using AAV-shRNA or scramble control, maintained on a MASH-inducing diet for 6 weeks. Mice were injected with CL-316,243 to stimulate lipolysis, and plasma glycerol levels were measured at the indicated time points. (n=7-9 mice per group, by Student’s *t* test). * *p* < 0.05, ** *p* < 0.01; bars indicate mean ± s.e.m..

5. The authors provide transcriptomics data suggesting decreased macrophage infiltration in the adipose deposits of the GPR146 KO mice. Is this supported by any other data such as histology?

Answer: As shown in the H&E staining of both iWAT and eWAT (new data, Fig. 2i, j) showed minimal immune cell infiltration in GPR146 KO mice, consistent with a lower inflammatory burden. In contrast, crown-like structures—indicative of macrophage accumulation around dead adipocytes—were occasionally observed in control mice. While we did not perform immunohistochemical staining for specific macrophage markers, the concordance between histological observations and transcriptomic signatures strengthens the conclusion that GPR146 deficiency is associated with reduced adipose tissue inflammation.

Fig.2

Figure 2. GPR146 Promotes HFD-induced Obesity in Mice. i,j, Representative images of (H&E)-stained eWAT with Crown-like structure (CLS) indicated by red arrows (i). Quantification of CLSs per 100 adipocytes (j). * *p* < 0.05, ** *p* < 0.01; bars in indicate mean ± s.e.m.

6. The study includes several types of omics data; however, the full data has not been made available and no repository for data storage has been indicated?

Answer: We have deposited the microarray data in GEO and are currently awaiting the assignment of an accession number, which will be provided once available.

References:

- 1 Kaikaew, K., Grefhorst, A. & Visser, J. A. Sex Differences in Brown Adipose Tissue Function: Sex Hormones, Glucocorticoids, and Their Crosstalk. *Front Endocrinol (Lausanne)* **12**, 652444 (2021). <https://doi.org/10.3389/fendo.2021.652444>
- 2 Goossens, G. H., Jocken, J. W. E. & Blaak, E. E. Sexual dimorphism in cardiometabolic health: the role of adipose tissue, muscle and liver. *Nat Rev Endocrinol* **17**, 47-66 (2021). <https://doi.org/10.1038/s41574-020-00431-8>
- 3 Stincone, A. *et al.* The return of metabolism: biochemistry and physiology of the pentose phosphate pathway. *Biol Rev Camb Philos Soc* **90**, 927-963 (2015). <https://doi.org/10.1111/brv.12140>
- 4 Chen, Z., Tian, R., She, Z., Cai, J. & Li, H. Role of oxidative stress in the pathogenesis of nonalcoholic fatty liver disease. *Free Radic Biol Med* **152**, 116-141 (2020). <https://doi.org/10.1016/j.freeradbiomed.2020.02.025>
- 5 Hu, X. *et al.* A gut-derived hormone regulates cholesterol metabolism. *Cell* **187**, 1685-1700 e1618 (2024). <https://doi.org/10.1016/j.cell.2024.02.024>
- 6 Wang, G. *et al.* Regulation of UCP1 and Mitochondrial Metabolism in Brown Adipose Tissue by Reversible Succinylation. *Mol Cell* **74**, 844-857 e847 (2019). <https://doi.org/10.1016/j.molcel.2019.03.021>
- 7 Chouchani, E. T. *et al.* Mitochondrial ROS regulate thermogenic energy expenditure and sulfenylation of UCP1. *Nature* **532**, 112-116 (2016). <https://doi.org/10.1038/nature17399>
- 8 Zhong, J. *et al.* adiposetissue.org: A knowledge portal integrating clinical and experimental data from human adipose tissue. *Cell Metab* **37**, 566-569 (2025). <https://doi.org/10.1016/j.cmet.2025.01.012>
- 9 Miranda, A. M. A. *et al.* Selective remodelling of the adipose niche in obesity and weight loss. *Nature* **644**, 769-779 (2025). <https://doi.org/10.1038/s41586-025-09233-2>
- 10 Bost, F., Aouadi, M., Caron, L. & Binetruy, B. The role of MAPKs in adipocyte differentiation and obesity. *Biochimie* **87**, 51-56 (2005). <https://doi.org/10.1016/j.biochi.2004.10.018>
- 11 Yu, H. *et al.* GPR146 Deficiency Protects against Hypercholesterolemia and Atherosclerosis. *Cell* **179**, 1276-1288 e1214 (2019). <https://doi.org/10.1016/j.cell.2019.10.034>
- 12 Barella, L. F., Jain, S., Kimura, T. & Pydi, S. P. Metabolic roles of G protein-coupled receptor signaling in obesity and type 2 diabetes. *FEBS J* **288**, 2622-2644 (2021). <https://doi.org/10.1111/febs.15800>
- 13 Wang, X. *et al.* Hepatocyte TAZ/WWTR1 Promotes Inflammation and Fibrosis in Nonalcoholic Steatohepatitis. *Cell Metab* **24**, 848-862 (2016). <https://doi.org/10.1016/j.cmet.2016.09.016>
- 14 Rimbart, A. *et al.* Variants in the GPR146 Gene Are Associated With a Favorable Cardiometabolic Risk Profile. *Arterioscler Thromb Vasc Biol* **42**, 1262-1271 (2022). <https://doi.org/10.1161/ATVBAHA.122.317514>

- 15 Alquier, T. & Poitout, V. Considerations and guidelines for mouse metabolic phenotyping in diabetes research. *Diabetologia* **61**, 526-538 (2018). <https://doi.org/10.1007/s00125-017-4495-9>
- 16 van Dijk, T. H. *et al.* A novel approach to monitor glucose metabolism using stable isotopically labelled glucose in longitudinal studies in mice. *Lab Anim* **47**, 79-88 (2013). <https://doi.org/10.1177/0023677212473714>

Point-by-Point Response

We sincerely thank the reviewers for their thoughtful evaluation of our revised manuscript and for the constructive comments provided. In response, we have carefully revised the manuscript and addressed all remaining concerns with additional data, new analyses, and further clarifications where appropriate.

REVIEWER COMMENTS

Reviewer #1 (Remarks to the Author):

I want to commend the authors are giving a strong rigorous effort to addressing my concerns and questions. Overall, I am quite satisfied. A few points remain for clarification:

-What is happening to the RER at the end of the Supp Figure 7 panel g?

Answer: We thank the reviewer for pointing this out. We also noticed the unusual increase in RER during the final 12-hour interval of the measurement (84-96 h in Supplementary Fig. 7g). We thoroughly investigated this by checking with the metabolic core facility, and they confirmed that no technical issues occurred during the recording period. For additional transparency, we now provide the raw VO_2 and VCO_2 traces (Figure a and b), which show consistent patterns, and no abrupt spikes were detected in locomotor or ambulatory activity (Figure c and d), suggesting that the RER elevation was not due to increased exploratory behaviour or physical activity.

Importantly, excluding the final 12-h interval does not affect our main conclusion: Adipoq-Cre⁺ mice exhibit a significantly higher RER than controls during the light (resting) phase, which remains robust even without this final segment. We have clarified this point in the revised figure legend and ensured that this anomalous time point does not bias the interpretation of our data.

Figure (a-d): Oxygen consumption (a), carbon dioxide production (b), locomotor activity (c) and Ambulatory activity (d) measured by indirect calorimetry in Adipoq-Cre⁺ and control mice (Adipoq-Cre⁻) fed HFD for 4 months.

-Quantification of the western blots would be beneficial

Answer: Thank you very much for this valuable suggestion. In response, we have performed densitometric quantification of the relevant Western blots and included the quantified data in the revised figures and figure legends. These additions strengthen the conclusions drawn from the immunoblotting results and provide clearer comparisons between experimental groups.

-Figure 3g-i has the EE for the females corrected by bodyweight. This should be changed to the data without bodyweight correction for accurate representation like the male data in the first part of Figure 3.

[\(https://pubmed.ncbi.nlm.nih.gov/40993210/\)](https://pubmed.ncbi.nlm.nih.gov/40993210/)

Answer: Thank you for this suggestion and for sharing the consensus guide. We have now re-analysed the data using the same CalR method as in Figure 3, and have replaced the original panels (Fig.3f,h, and i).

Fig. 3g, Indirect calorimetry study in female mice fed HFD for two weeks. Hourly energy expenditure (EE) in female *Gpr146*^{+/+} and *Gpr146*^{-/-} mice was recorded over 60 hours (n=8 mice per group). **3h,** Mean hourly energy expenditure (kcal/hour) per mouse during total, dark, and light phases. * p< 0.05, ** p< 0.01

-Following up on the EE, I suppose the adipose KO mice in indirect calorimetry, paired with the cell data on isolated adipocytes, would suggest the EE is not arising from the fat tissue?

Answer: We thank the reviewer for the important follow-up question. The combined data from our indirect calorimetry studies of adipose KO and Seahorse assays indeed suggest that the increased energy expenditure (EE) observed in whole-body *Gpr146* knockout mice, particularly in males, **is unlikely to originate solely from white adipose tissue.**

However, the lack of a measurable EE increase in adipose-specific *Gpr146* knockout mice does not fully exclude an adipose contribution to the elevated EE observed in whole-body knockouts. Two factors may help explain this discrepancy. First, Adipoq-Cre-mediated deletion of GPR146 is attenuated in visceral fat depots under prolonged

HFD feeding. This partial depletion likely contributes to the more modest reductions in body weight and adiposity observed in adipose-specific knockouts compared to global knockouts. Second, our data suggest that GPR146 regulates both preadipocyte differentiation and lipolysis in mature adipocytes. Since Adipoq-Cre is active mainly in mature adipocytes, the model may not capture the full functional repertoire of GPR146 in adipose tissue. As a result, the partial phenotypic effects in adipose-specific knockouts may be insufficient to elicit a detectable increase in energy expenditure by indirect calorimetry.

Our results also reveal a sex-specific divergence in thermogenic responses. In female *Gpr146*^{-/-} mice, we observed a robust (>15-fold) induction of *Ucp1* in eWAT, accompanied by upregulation of *Pgc1α* and *Ppara*, indicating potential activation of UCP1-mediated thermogenesis. In contrast, male knockouts showed no transcriptional induction of *Ucp1* or other classical thermogenic markers, and neither creatine nor calcium futile cycling pathways appeared upregulated at the gene expression level.

Taken together, these data suggest that while adipose tissue-resident GPR146 plays an important role in regulating adipogenesis and lipid mobilization, adipose tissue is unlikely to be the sole driver of the elevated energy expenditure- particularly in males. The precise mechanisms underlying this phenotype remain unresolved and are likely to involve systemic factors, non-adipose tissue contributions, or inter-organ crosstalk. **We appreciate the reviewer raising this point, and we have honestly acknowledged that the detailed molecular mechanism by which GPR146 deficiency enhances energy expenditure remains unclear in male mice and requires further investigation.**